# Cdc4 phospho-degrons allow differential regulation of Ame1[CENP-U] protein stability across the cell cycle

Miriam Böhm[1], Kerstin Killinger[1], Alexander Dudziak[1], Pradeep Pant[2], Karolin Jänen[1], Simone Hohoff[1], Karl Mechtler[3,4,5], Mihkel Örd[6], Mart Loog[6], Elsa Sanchez-Garcia[2], Stefan Westermann[1]*

[1]Department of Molecular Genetics I, Faculty of Biology, Center of Medical Biotechnology, University of Duisburg-Essen, Essen, Germany; [2]Department of Computational Biochemistry, Faculty of Biology, Center of Medical Biotechnology, University of Duisburg-Essen, Essen, Germany; [3]IMP - Research Institute of Molecular Pathology, Vienna, Austria; [4]Institute of Molecular Biotechnology of the Austrian Academy of Sciences (IMBA), Vienna Biocenter (VBC), Vienna, Austria; [5]Gregor Mendel Institute (GMI), Austrian Academy of Sciences, Vienna BioCenter (VBC), Vienna, Austria; [6]Institute of Technology, University of Tartu, Tartu, Estonia

**Abstract** Kinetochores are multi-subunit protein assemblies that link chromosomes to microtubules of the mitotic and meiotic spindle. It is still poorly understood how efficient, centromere-dependent kinetochore assembly is accomplished from hundreds of individual protein building blocks in a cell cycle-dependent manner. Here, by combining comprehensive phosphorylation analysis of native Ctf19[CCAN] subunits with biochemical and functional assays in the model system budding yeast, we demonstrate that Cdk1 phosphorylation activates phospho-degrons on the essential subunit Ame1[CENP-U], which are recognized by the E3 ubiquitin ligase complex SCF-Cdc4. Gradual phosphorylation of degron motifs culminates in M-phase and targets the protein for degradation. Binding of the Mtw1[Mis12] complex shields the proximal phospho-degron, protecting kinetochore-bound Ame1 from the degradation machinery. Artificially increasing degron strength partially suppresses the temperature sensitivity of a *cdc4* mutant, while overexpression of Ame1-Okp1 is toxic in SCF mutants, demonstrating the physiological importance of this mechanism. We propose that phospho-regulated clearance of excess CCAN subunits facilitates efficient centromere-dependent kinetochore assembly. Our results suggest a novel strategy for how phospho-degrons can be used to regulate the assembly of multi-subunit complexes.

*For correspondence:
Stefan.Westermann@uni-due.de

Competing interests: The authors declare that no competing interests exist.

## Introduction

Kinetochores form a dynamically regulated binding interface between chromosomes and the mitotic spindle, a connection that is essential to partition the genetic material equally during cell division (*Biggins, 2013*). As chromatin-bound multi-protein complexes, kinetochores assemble exclusively at centromeres. Biochemical experiments have elucidated the connectivity between multiple subcomplexes in the context of an assembled kinetochore (*Musacchio and Desai, 2017*), and structural biology has started to define the organization of these building blocks at atomic resolution (*Hinshaw and Harrison, 2019*; *Yan et al., 2019*). It is, however, still poorly understood how kinetochore proteins first assemble into stable subcomplexes and then associate into higher-order structures with the correct stoichiometry specifically at the centromere.

Budding yeast centromeres are attached to kinetochore microtubules during almost the entire vegetative cell cycle. Upon replication, kinetochores assemble quickly on both sister chromatids to ensure effective bi-orientation. Live cell imaging experiments suggest that kinetochore assembly upon replication of centromere DNA occurs rapidly and is completed within 10–15 min (*Tanaka et al., 2007*). This is remarkable as even the relatively simple kinetochores in budding yeast consist of more than 40 different proteins. These protein subunits assemble in two layers: the inner kinetochore comprises 16 subunits, which are collectively called the constitutive centromere-associated network (CCAN, or Ctf19$^{CCAN}$ in budding yeast). The CCAN contains the essential protein Mif2$^{CENP-C}$, which contacts and organizes many other subunits of the CCAN (*Klare et al., 2015*) and, among other subcomplexes, the COMA complex, which consists of the conserved subunits Ctf19$^{CENP-P}$, Okp1$^{CENP-Q}$, Mcm21$^{CENP-O}$, and Ame1$^{CENP-U}$, of which Ame1 and Okp1 are essential for cell viability in budding yeast (*Hornung et al., 2014*). The CCAN directly binds to a specialized nucleosome by recognizing the histone H3 variant Cse4$^{CENP-A}$ via the Mif2 and Ame1-Okp1 subunits (*Anedchenko et al., 2019*; *Fischböck-Halwachs et al., 2019*; *Killinger et al., 2020*).

The outer kinetochore forms a microtubule-binding layer with the KMN network at its core. As this network lacks DNA-binding proteins, it relies on the CCAN for centromere recruitment. The Knl1c (Spc105c in yeast), Mis12c (Mtw1c in yeast), and Ndc80c subcomplexes are necessary to efficiently couple dynamic microtubules to centromeres (*Cheeseman et al., 2006*; *Musacchio and Desai, 2017*) and also form a regulated recruitment platform for components of the spindle assembly checkpoint (SAC) (*Joglekar, 2016*; *London et al., 2012*). The outer yeast kinetochore also includes a ring-forming protein complex called the Dam1c, which is required for robust coupling of the KMN network to dynamic microtubules (*Lampert et al., 2013*; *Tien et al., 2010*).

A number of control mechanisms for kinetochore assembly have been described, and the best studied examples have defined the regulation of the histone H3 variant Cse4$^{CENP-A}$. High levels of Cse4$^{CENP-A}$ lead to its mis-incorporation into non-centromeric loci (*Heun et al., 2006*; *Ranjitkar et al., 2010*; *Van Hooser et al., 2001*) and can promote the formation of ectopic kinetochores and genetic instability (*Amato et al., 2009*; *Hildebrand and Biggins, 2016*). In budding yeast, Cse4$^{CENP-A}$ protein levels are controlled via regulated proteolysis involving the E3 ubiquitin ligase Psh1, which prevents mis-incorporation of Cse4 into chromosome arms (*Ranjitkar et al., 2010*). Additional links between the ubiquitin-proteosome system and kinetochore assembly in budding yeast are provided by the Mub1/Ubr2 E3 ligase complex, which regulates the level of the Dsn1 subunit (*Akiyoshi et al., 2013*). Whether similar regulatory systems operate for other kinetochore subunits and how they may contribute to kinetochore assembly is currently not known.

Phospho-regulation plays an important role for multiple aspects of kinetochore function. The Ipl1$^{AuroraB}$ kinase regulates kinetochore-microtubule attachments during error correction to ensure sister chromatid bi-orientation, and it also promotes kinetochore assembly by phosphorylating the Mis12c subunit Dsn1 (*Dimitrova et al., 2016*). The major regulator of the mitotic cell cycle Cdc28$^{Cdk1}$, on the other hand, promotes outer kinetochore assembly in human cells by stimulating the phospho-dependent recruitment of Ndc80 via CENP-T (*Huis In 't Veld et al., 2016*; *Nishino et al., 2013*). By contrast, in budding yeast the Cnn1-Ndc80c interaction is negatively regulated by Mps1 phosphorylation (*Malvezzi et al., 2013*). Apart from these examples, however, it is still unclear how phospho-regulation is linked to cell cycle progression and which aspects of kinetochore function may be affected.

Here we set out to define mechanisms of phospho-regulation in the context of the budding yeast inner kinetochore. We identify multiple Cdk1 substrates in the yeast Ctf19$^{CCAN}$ complex and demonstrate that a subset of these sites constitute phospho-degron motifs that are activated in a cell cycle-dependent manner and recognized by the SCF-Cdc4 E3 ubiquitin ligase complex. Our results reveal a molecular link between the core cell cycle machinery and key protein subunits of the inner kinetochore that serves to couple kinetochore subunit turnover to cell cycle progression.

**Table 1.** Analysis of Ctf19$^{CCAN}$ phosphorylation in yeast extracts.

Native constitutive centromere-associated network (CCAN) phosphorylation sites detected after purification of TAP-tagged kineto-chore subunits from yeast extracts. For details, see *Figure 1—source data 1*.

| *S.c.*CCAN subunit | Human homolog | % sequence coverage | Total P-sites detected | Minimal Cdk1 sites detected (S/T)P | Full Cdk1 sites detected (S/TP_K/R) |
|---|---|---|---|---|---|
| Ame1 | CENP-U | 79 | 8 | 4 (T31, S41, S45, S53) | - |
| Okp1 | CENP-Q | 85 | 6 | - | - |
| Mcm21 | CENP-O | 92 | 3 | - | 1 (S139) |
| Ctf19 | CENP-P | 80 | 1 | - | - |
| Nkp1 | - | 97 | 3 | - | 1 (S222) |
| Nkp2 | - | 89 | - | - | |
| Chl4 | CENP-N | 98 | 3 | 1 (S281) | - |
| Iml3 | CENP-L | 95 | - | - | - |
| Ctf3 | CENP-I | 75 | - | - | - |
| Mcm22 | CENP-H | 97 | - | - | - |
| Mcm16 | CENP-K | 93 | - | - | - |
| Cnn1 | CENP-T | 90 | 17 | 2 (T42, S192) | 2 (T21, S177) |
| Mhf1 | CENP-S | 93 | 3 | 1 (T34) | - |
| Mhf2 | CENP-X | 95 | 1 | 1 (S60) | - |

# Results

## Mapping of native phosphorylation sites identifies multiple candidate Cdk1 substrates in the yeast CCAN

To investigate the phosphorylation status of native yeast CCAN subunits, we affinity-purified TAP-tagged components representing major CCAN subcomplexes (Chl4$^{CENP-N}$-TAP, Mcm16$^{CENP-H}$-TAP, Cnn1$^{CENP-T}$-TAP, Mhf1$^{CENP-S}$-TAP, Mhf2$^{CENP-X}$-TAP) from log-phase yeast extracts and identified phosphorylation sites by mass spectrometry. In total, this analysis detected more than 70 phosphorylation sites on nine different CCAN subunits (*Table 1*, *Figure 1—source data 1*). Of these sites, nine followed the minimal consensus motif for Cdk1 phosphorylation (S/TP), while four sites (in the subunits Mcm21$^{CENP-O}$, Nkp1, and Cnn1$^{CENP-T}$) followed the full Cdk1 consensus (S/TP_K/R). While Cdk1 phosphorylation of Cnn1 has been described before (*Bock et al., 2012*; *Schleiffer et al., 2012*), our analysis identified Ame1 as a candidate Cdk1 target, with a cluster of four minimal Cdk1 sites being phosphorylated in the N-terminus of the protein.

## The essential CCAN subunit Ame1$^{CENP-U}$ is a Cdk1 substrate in vivo and in vitro

Since Ame1 was the only essential protein among the candidate Cdk1 substrates and the cluster of Cdk1 phosphorylation sites is located close to the Mtw1c binding domain, we focused our analysis on this subunit (*Figure 1A*). We phosphorylated recombinant Ame1-Okp1 complex (AOc) with purified Cdc28-Clb2 (M-Cdk1) or Cdc28-Clb5 (S-Cdk1) in vitro and mapped phosphorylation sites by mass spectrometry (*Figure 1—source data 2*). Ame1 displayed a noticeable shift in migration in SDS-PAGE upon phosphorylation with Cdc28-Clb2, but less so with Cdc28-Clb5 (*Figure 1B*). Quantitative phosphorylation analysis confirmed that in the Ame1-Okp1 complex Clb2-Cdk1 preferentially phosphorylated Ame1, whereas Clb5-Cdk1 preferred Okp1. The phosphorylation of the Ame1-Okp1 complex was dependent on the hydrophobic patch, a known substrate docking region in Clb5 and Clb2 cyclins, and a slightly stronger docking potentiation was seen in case of Okp1 (*Figure 1—figure supplement 1A*).

The mapped phosphorylation sites closely corresponded to the sites detected on native Ame1, in particular phosphorylation of the residues Thr31, Ser41, Ser45, and Ser52/Ser53 was both detected in vivo and in vitro (*Figure 1C*). In case of Ser52/Ser53 either one, or both adjacent sites may be phosphorylated. Conservation of these phospho-sites can be detected in the most closely related

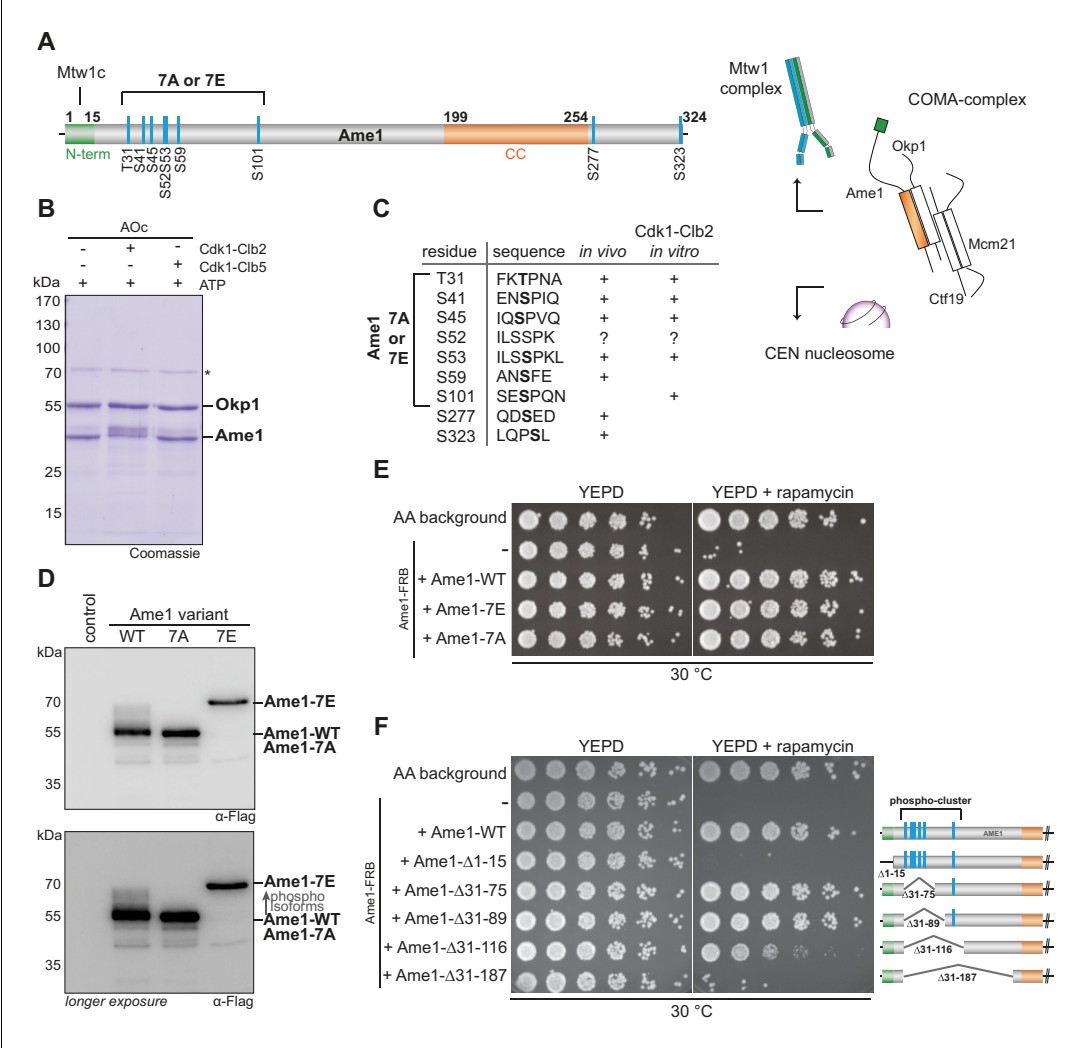

**Figure 1.** Phosphorylation analysis of the essential constitutive centromere-associated network (CCAN) subunit Ame1. (A) Organization of the essential CCAN component Ame1[CENP-U] and localization of phosphorylation sites. Ame1 shows a Cdk1 phosphorylation cluster (T31, S41, S45, S52, S53, S59, S101) at the N-terminus. The first 15 amino acids are essential for Mtw1c binding, the coiled-coil region (aa 199–254) is required for heterodimerization with Okp1[CENP-Q]. Schematic overview on the right shows the four-protein complex COMA, consisting of Ame1[CENP-U], Okp1[CENP-Q], Ctf19[CENP-P], and Mcm21[CENP-O]. The COMA complex binds to the outer kinetochore component Mtw1 complex and to the centromeric nucleosome. (B) In vitro kinase assay with recombinant Ame1-Okp1c with either Cdk1-Clb2 or Cdk1-Clb5. The migration pattern of Ame1 is shifted to a slowly migrating form when incubated with Cdk1-Clb2. Asterisk denotes a contaminating protein. (C) List of all mapped Ame1 phosphorylation sites either in vivo or in vitro. T31, S41, S45, S53, and S101 show the minimal motif for Cdk1 (S/TP). (D) Stably integrated Ame1 variants display distinct migration patterns in SDS-PAGE. Ame1-WT shows multiple slowly migrating forms that are eliminated in Ame1-7A and Ame1-7E. (E) Serial dilution assay of Ame1 variants using the FRB anchor-away system. Ame1-WT and both mutants can rescue the growth defect when endogenous Ame1 is anchored away from the nucleus. (F) Serial dilution assay of internal Ame1 truncation mutants in the anchor-away system.

The online version of this article includes the following source data and figure supplement(s) for figure 1:

**Source data 1.** Mass spectrometry analysis of native constitutive centromere-associated network (CCAN) complexes.

**Source data 2.** Mass spectrometry analysis of in vitro phosphorylated COMA.

**Figure supplement 1.** Quantitative phosphorylation analysis of recombinant Ame1-Okp1 by S-Cdk1 and M-Cdk1 complexes.

*Saccharomyces* species (*Figure 1—figure supplement 1B*). For the subsequent analysis, we focused on seven N-terminal phosphorylation sites as the two C-terminal sites Ser277 and Ser323 were not phosphorylated by Cdc28-Clb2 in vitro. To analyze the functional role of Ame1 phosphorylation, we mutated the cluster to either alanine (Ame1-7A) or glutamic acid (Ame1-7E) to eliminate or mimic phosphorylation, respectively. We integrated Flag-tagged Ame1 constructs under their endogenous

promoter with these mutations into yeast and analyzed cell extracts by western blotting. Analysis of log phase extracts showed that wild-type Ame1 displayed multiple slowly migrating forms that were eliminated in the 7A mutant (*Figure 1D*). By contrast, Ame1-7E migrated much more slowly than wild-type, its position in SDS-PAGE corresponding to the most slowly migrating forms of Ame1-WT. Ame1-7A and -7E mutants were viable when expressed as the sole source of Ame1 in the cell. In an anchor-away approach, in which endogenous Ame1 is removed from the nucleus upon addition of rapamycin, both Ame1-7A and -7E variants supported viability with little difference in growth rate on rich media compared to wild-type Ame1 (*Figure 1E*). Analysis of internal truncations, which maintained the essential Mtw1c-binding N-terminus (residues 1–15), showed that deleting the region harboring the entire phospho-cluster (Δ31–116) yielded a slow growth phenotype, while a more extensive deletion was inviable (*Figure 1F*). We conclude that Ame1 phosphorylation is not required for viability, but the N-terminus contributes to an important aspect of Ame1 function, even when the Mtw1-binding domain is retained. This function may either be sequence-specific, lie in the correct positioning of the N-terminus, or involve a combination of both of these aspects.

## Non-phosphorylatable Ame1 mutants accumulate to increased protein levels

During our cellular characterization experiments for Ame1 phospho-mutants, we expressed wild-type or mutant versions of Ame1, along with its binding partner Okp1 from a two-micron plasmid under control of a galactose-inducible promoter (*Figure 2A*). Western blot analysis showed that wild-type Ame1 gradually accumulated over the course of 5 hr after switching the cells to galactose. Strikingly, the non-phosphorylatable Ame1-7A mutant accumulated to much higher protein levels in the same time span, leading to a roughly fourfold increase in steady-state level compared to wild-type after 5 hr in galactose (*Figure 2B, C*). In this experiment, the Ame1-7E mutant behaved similar to the -7A mutant, suggesting that it may constitute a phospho-preventing rather than a phospho-mimetic mutation (*Figure 2B*). By contrast, Okp1 expressed from the same plasmid showed no change in protein level in the different Ame1 mutants, arguing that differences in plasmid stability or mitotic retention cannot be the cause for the observed effect on the Ame1 protein level. In order to simplify quantification of the Ame1 signal, whole-cell extracts were treated with lambda phosphatase. This resulted in a collapse of the slowly migrating forms of Ame1-WT. Quantification of protein levels after phosphatase treatment confirmed a fourfold increase in the steady-state level of Ame1-7A in the wild-type strain background (*Figure 2—figure supplement 1A*).

The mitotic checkpoint delays anaphase onset until all kinetochores have achieved correct attachment to the spindle. We analyzed overexpression of Ame1-Okp1 wild-type or phosphorylation mutants in strains lacking the checkpoint component *mad1*. This mutant should eliminate mitotic delays that could in principle contribute to protein level differences. This experiment confirmed the accumulation of Ame1 phosphorylation mutants (*Figure 2—figure supplement 1C*). As the steady-state protein level is determined by the rate of protein translation versus degradation, and the rate of production should be unaffected in these experiments, we reasoned that non-phosphorylatable Ame1 mutants may accumulate due to impaired protein degradation. The levels of Cse4, part of the centromeric nucleosome and a direct binding partner of Ame1-Okp1, have been shown to be regulated by ubiquitin-dependent proteolysis via the E3 ubiquitin ligase Psh1 (*Ranjitkar et al., 2010*). Levels of GAL-overexpressed Ame1-WT, however, remained low in a *psh1Δ* strain background, while Ame1-7A and -7E accumulated as in the wild-type background (*Figure 2—figure supplement 1D*). This suggests that Psh1 is not involved in Ame1 level regulation under these conditions. Another E3 ubiquitin ligase complex, Ubr2/Mub1 has been shown to regulate Dsn1, which is a subunit of the Ame1 binding partner Mtw1c (*Akiyoshi et al., 2013*). Similar to *psh1Δ*, however, Ame1 protein levels were unaffected by the *mub1* deletion, and we conclude that Ubr2/Mub1 is not involved in Ame1 level regulation under these conditions either (*Figure 2—figure supplement 1E*).

## Identification of two phospho-degron motifs in the Ame1 N-terminus

To delineate the contribution of individual phosphorylation sites to Ame1 protein level regulation in the overexpression setting, we constructed mutants in which we prevented phosphorylation at selected sites individually or in combination (*Figure 2D*). Analysis of Ame1 protein levels after 5 hr of expression in the presence of galactose showed that preventing phosphorylation on Thr31 had

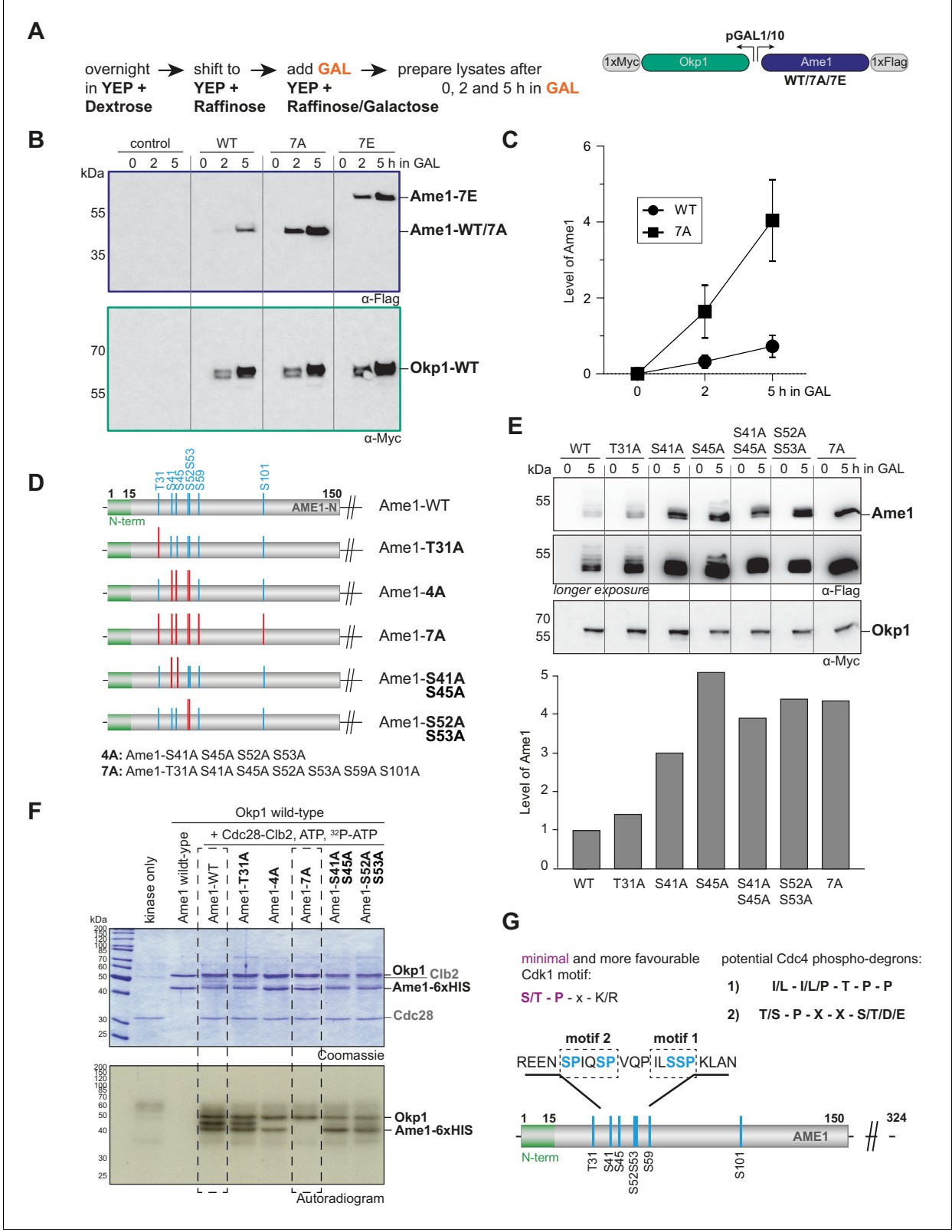

**Figure 2.** Identification of phospho-degron motifs in Ame1. (**A**) Flag- and Myc-tagged versions of Ame1 and Okp1 were expressed from a two-micron plasmid under a bidirectional galactose-inducible promoter. Under normal growth conditions in YEP + dextrose or YEP + raffinose, no overexpression occurs, overexpression is only induced by adding galactose (GAL) to the medium. After 0, 2, and 5 hr in GAL, cell extracts were prepared and protein expression was followed using western blot analysis. (**B**) Western blot analysis of overexpressed Ame1-WT, -7A, and -7E variants in a wild-type strain

*Figure 2 continued on next page*

*Figure 2 continued*

background. (**C**) Quantification of protein levels of Ame1-WT and Ame1-7A after indicated times in galactose medium. Mean values and standard error of the mean are indicated, n = 7. (**D**) Overview of Ame1 phospho-mutants used for overexpression studies (**E**) or in vitro kinase assays (**F**). (**E**) Overexpression studies of individual Ame1 phospho-variants. Ame1 protein levels in this experiment are quantified below, the Ame1-WT level is set to 1. Okp1 levels are stable and used for normalization of Ame1 protein levels. (**F**) In vitro kinase assay of AO complexes using recombinant Cdk1-Clb2. Note reduced or lacking phosphorylation of Ame1-4A and -7A, respectively. Also Okp1 can be phosphorylated by Cdk1-Clb2. (**G**) Cdk1 target sites in Ame1 resemble two different types of phospho-degrons motifs that are recognized by the E3 ubiquitin ligase complex SCF-Cdc4.

The online version of this article includes the following figure supplement(s) for figure 2:

**Figure supplement 1.** Additional analysis of Ame1 overexpression.
**Figure supplement 2.** Additional analysis of AO in vitro phosphorylation by Cdc28-Clb2.

relatively little effect on Ame1 level when compared to the wild-type. By contrast, preventing phosphorylation at Ser41, Ser45, or Ser52/53 led to accumulation of the protein, roughly similar to preventing phosphorylation altogether in the 7A mutant (*Figure 2E*). We prepared the analogous Ame1 mutants as recombinant Ame1-Okp1 (AO) complexes for in vitro kinase assays to evaluate the contribution of these individual sites to overall Ame1 phosphorylation. Autoradiographs showed that in addition to Ame1 also Okp1 can be phosphorylated by Cdc28-Clb2 (*Figure 2F*, see also *Figure 1—figure supplement 1* and *Figure 2—figure supplement 2*). Ame1-WT appeared as two separated phosphorylated forms after in vitro phosphorylation. The Ame1-T31A mutant displayed a similar phosphorylation pattern, while the phosphorylation of Ame1-4A was clearly decreased, with only the fast migrating Ame1 form remaining. The Ame1-7A mutant completely eliminated Cdc28-Clb2 phosphorylation in vitro. Preventing phosphorylation on either Ser41/Ser45 or Ser52/Ser53 allowed some residual phosphorylation, but clearly decreased phosphorylation compared to wild-type. We conclude that the residues responsible for Ame1 level regulation in vivo are major targets for Cdc28 phosphorylation in vitro. Further analysis confirmed that mutating the candidate Cdk1 site Ser26 in Okp1 to alanine prevented Cdc28 phosphorylation and that the Ame1-7A/Okp1-1A complex was completely refractory to Cdc28 phosphorylation (*Figure 2—figure supplement 2*).

Post-translational modification via phosphorylation can be mechanistically linked to the control of protein stability via the generation of so-called phospho-degrons (*Skowyra et al., 1997*). The best studied example for this mechanism is the controlled ubiquitination and degradation of key cell cycle regulators by modular SCF complexes, using F-box proteins as readers of phosphorylated substrates (*Feldman et al., 1997*; *Örd et al., 2019a*; *Örd et al., 2019b*). Intriguingly, the Ame1 N-terminal sequences resembled previously described Cdc4 phospho-degrons: the Ame1 sequence surrounding Ser52 and Ser53 (motif 1) showed similarity to a cyclin E-type phospho-degron, comprehensively described in the context of the Cdk1 inhibitor Sic1 (*Kõivomägi et al., 2011*; *Nash et al., 2001*), while the combination of Ser41/Ser45 (motif 2) resembled a di-phospho-degron with a typical +4 spacing between phosphorylated residues found for example in the acetyltransferase Eco1 (*Hao et al., 2007*; *Lyons et al., 2013*; *Lyons and Morgan, 2011*; *Figure 2G*).

## Molecular dynamics simulations predict Ame1 phospho-peptide binding to Cdc4

To evaluate candidate degron motifs in Ame1, we performed Gaussian-accelerated molecular dynamics (GaMD) simulations of Ame1 phospho-peptide binding to the WD40 domain of Cdc4 using as template for initial coordinates the published crystal structure of a cyclin E-derived model peptide associated with Cdc4 (*Orlicky et al., 2003*). The analysis of the trajectories allowed us to predict that the doubly phosphorylated Ser41/Ser45 peptide should be a good Cdc4 binder. Phospho-Ser41 establishes hydrogen bond interactions with the guanidino groups of Arg485 and Arg534 of Cdc4 while phospho-Ser45 of the peptide establishes hydrogen bond interactions with Arg467 of Cdc4 (*Figure 3A*, *Figure 3—figure supplement 1A*). Further, phospho-Ser45 is engaged in additional interactions with Arg443, Ser464, and Thr465 of Cdc4 (*Figure 3B*). The peptide residues Pro42, Ile43, Glu39, and Asn40 are also involved in interactions with the protein (*Figure 3B*). Overall, the doubly phosphorylated peptide showed a strong hydrogen bond network at the binding pocket of Cdc4, highlighting the potential of this peptide as a Cdc4 binder.

The simulations also indicated that the doubly phosphorylated Ser41/Ser45 peptide establish more interactions with Cdc4 with respect to the singly phosphorylated variants (*Figure 3—figure*

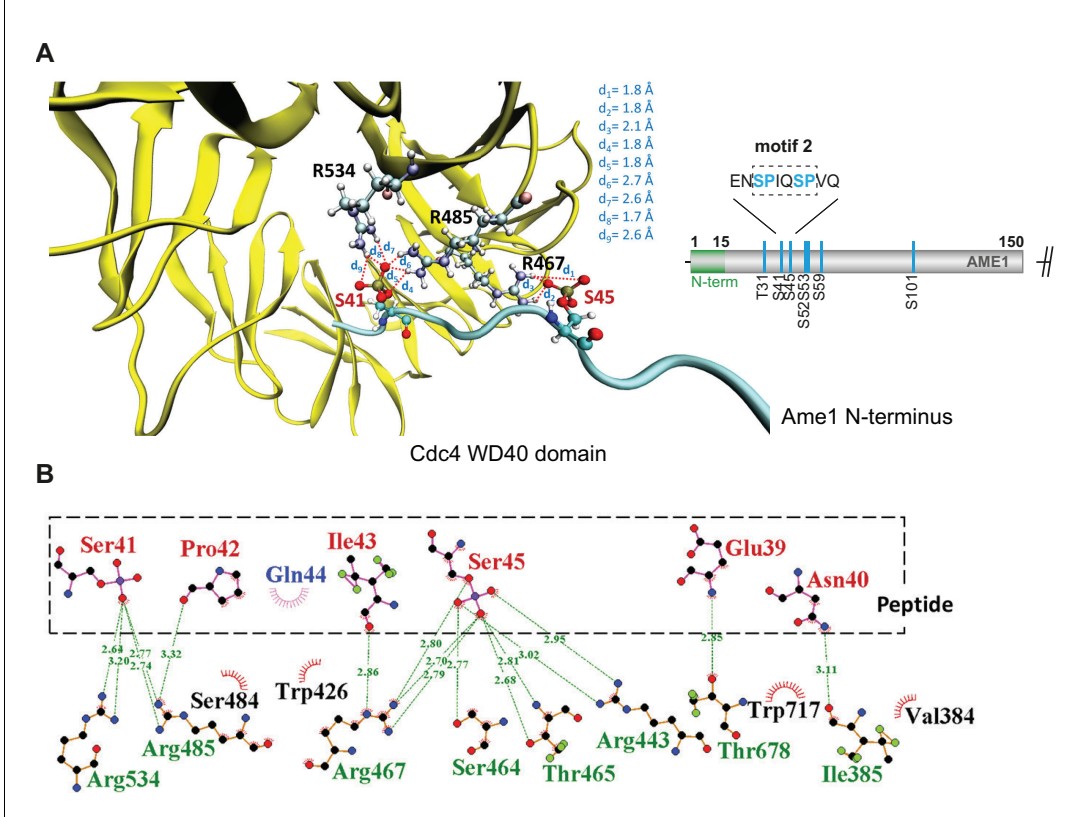

**Figure 3.** Gaussian-accelerated molecular dynamics simulations predict Ame1 peptide binding to Cdc4. (A) Interactions between the conserved arginine residues of Cdc4 (yellow) and the phospho-serine residues of the doubly phosphorylated peptide (cyan). (B) The doubly phosphorylated Ser41/Ser45 peptide and Cdc4 establish an intense hydrogen bond network involving the phosphorylated residues of the peptide and the conserved arginine residues of Cdc4, as well as other residues.

The online version of this article includes the following figure supplement(s) for figure 3:

**Figure supplement 1.** Additional analysis of peptide-Cdc4 interactions by Gaussian-accelerated molecular dynamics simulations.

supplement 1B, *Videos 1–3*). Furthermore, the protein-peptide complex involving the doubly phosphorylated peptide displayed less structural fluctuations during the trajectories compared to the simulations of Cdc4 with the monophosphorylated peptides, especially the Ser45 phosphorylated peptide. Accordingly, the doubly phosphorylated Ser41/Ser45 peptide remained attached at the binding site of the protein by establishing conserved interactions throughout the simulations, unlike the monophosphorylated peptides that displayed a dynamic behavior with less retention of their binding motifs (*Videos 1–3*).

## The SCF ligase with the F-box protein Cdc4 regulates Ame1-Okp1 protein levels

To test whether Ame1 level regulation is under control of SCF via phospho-degrons in vivo, we used different SCF mutant alleles in the GAL overexpression setting, starting with Skp1 as a component of all modular SCF complexes. All SCF *ts* alleles were used at the permissive temperature, ensuring progression through the cell cycle. Western blotting showed that Ame1-WT expressed from the GAL promoter strongly accumulated in the *skp1-3* mutant relative to a wild-type background (*Figure 4A*). The phospho-forms of Ame1 were preserved under these conditions, showing that the *skp1-3* mutant uncouples phosphorylation of Ame1 from its degradation. Interestingly, under these conditions, also an accumulation of Okp1, expressed from the same plasmid, was apparent. Okp1 appeared in two distinctly migrating forms, possibly corresponding to phosphorylation. Combining the Ame1-7A mutation with the *skp1-3* background revealed that the Ame1-7A protein was further enriched in the *skp1-3* background compared to the wild-type strain, demonstrating that Ame1-7A

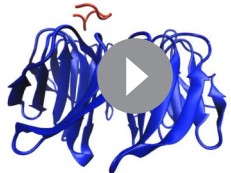

P. Pant & E. Sanchez-Garcia, Computational Biochemistry University of Duisburg-Essen

**Video 1.** Gaussian-accelerated molecular dynamics (GaMD) simulation of the doubly phosphorylated (S41 and S45) Ame1 peptide (red) binding to WD40 domain of Cdc4 (blue).

https://elifesciences.org/articles/67390#video1

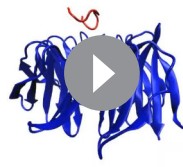

P. Pant & E. Sanchez-Garcia, Computational Biochemistry University of Duisburg-Essen

**Video 2.** Gaussian-accelerated molecular dynamics (GaMD) simulation of the monophosphorylated (S41) Ame1 peptide (red) binding to the WD40 domain of Cdc4 (blue).

https://elifesciences.org/articles/67390#video2

can be further accumulated by compromising the SCF machinery in addition to preventing phosphorylation of Ame1 itself (*Figure 2—figure supplement 1B*). We extended this analysis to mutant alleles in other SCF subunits, in particular to identify which F-box protein is responsible for Ame1 regulation. Similar to the *skp1-3* mutant, overexpressed Ame1 accumulated in mutant alleles of the Cullin subunit Cdc53[CUL1], the E2 enzyme Cdc34, and the F-box protein Cdc4. By contrast, Ame1 levels remained low (or were even decreased relative to wild-type) in a deletion mutant of the cytoplasmic F-box protein Grr1 (*Figure 4B*). Interestingly, the SCF mutants also had a pronounced effect on the level of overexpressed Okp1, with particularly strong accumulation (30-fold increase) observable in the *cdc4-1* mutant. In the background of the *cdc34-2* allele, Okp1 accumulated only slightly when Ame1 was wild-type, but more strongly when phosphorylation of Ame1 was prevented (*Figure 4B*). This indicates that in the context of the Ame1-Okp1 complex, level regulation by phosphorylation may occur both in cis (only affecting the subunit itself) or in trans (affecting also an interaction partner).

We tested the effect of Ame1-Okp1 expression from a GAL promoter in a serial dilution assay. In a wild-type strain background, AO overexpression was tolerated well. In a *skp1-3* mutant background, however, overexpression of AO, either in wild-type form or with Cdk1 sites mutated to alanine, compromised growth at 30 and 34°C (*Figure 4C*). Similar results were obtained for the *cdc34-2* mutant background, in which overexpression of AO already greatly impaired growth at 30°C. These effects are consistent with AO being physiological substrates of the SCF machinery, and they show that accumulation of AO can negatively impact cell growth. While overexpression of Ame1-Okp1 or Ctf19-Mcm21 (CM) subcomplexes was tolerated in a wild-type strain background, overexpression of the full four-protein COMA complex was toxic, indicating that maintenance of the proper protein level of this complex is critical for viability (*Figure 4D*).

### Increasing degron strength in the Ame1 N-terminus partially suppresses a *cdc4* mutant

To further study the regulation of Ame1-Okp1 by phospho-degrons, we converted the motif 1 degron (ILSSP) of Ame1 into a stronger Cdc4 phospho-degron (ILTPP), which was shown in the context of the Cdk1 inhibitor Sic1 to provide the highest binding affinity for Cdc4 (Ame1-CPD[ILTPP]; *Nash et al., 2001*). GaMD simulations were performed to study the binding of this Ame1-derived peptide featuring a

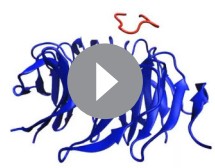

P. Pant & E. Sanchez-Garcia, Computational Biochemistry University of Duisburg-Essen

**Video 3.** Gaussian-accelerated molecular dynamics (GaMD) simulation of the monophosphorylated (S45) Ame1 peptide (red) binding to the WD40 domain of Cdc4 (blue).

https://elifesciences.org/articles/67390#video3

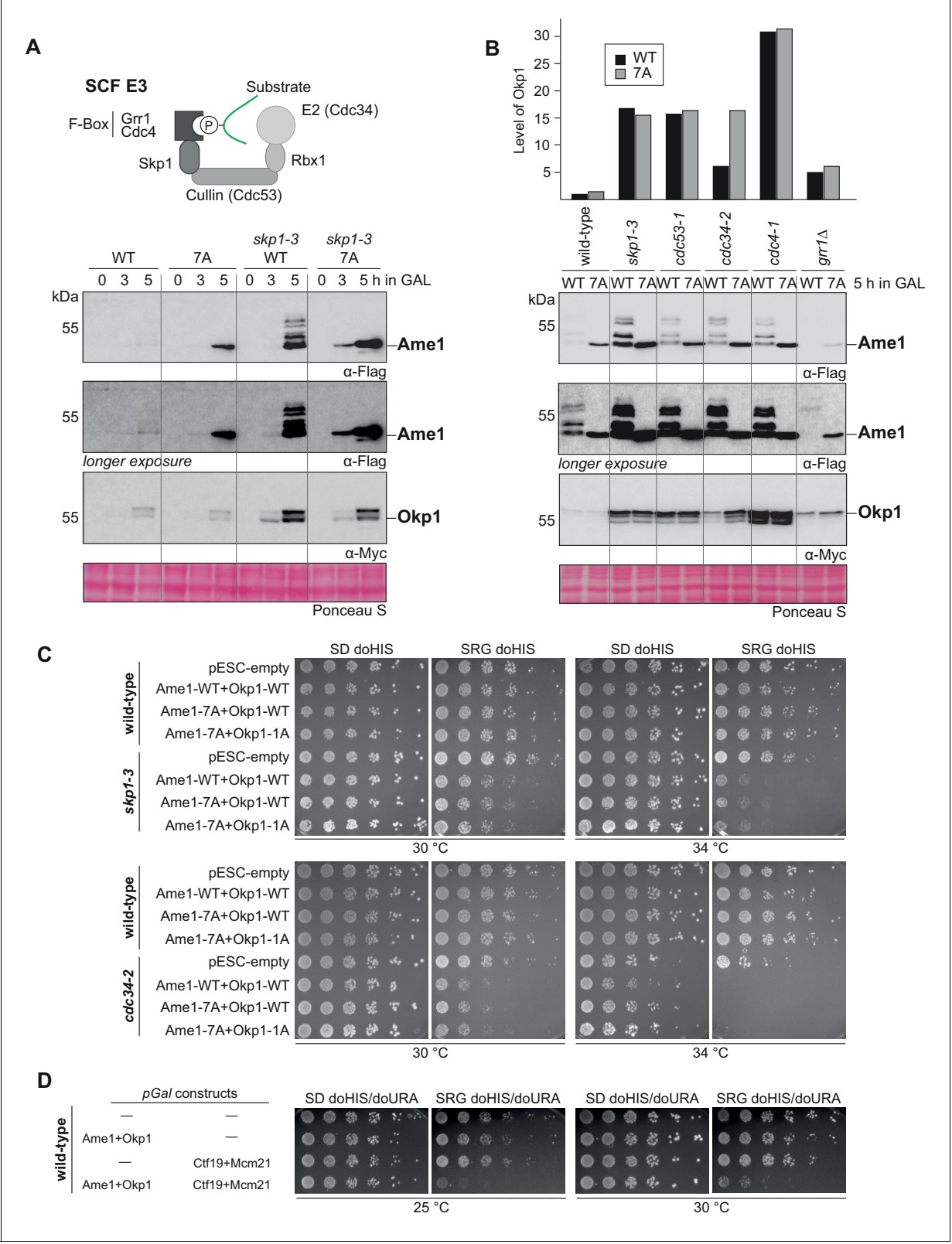

**Figure 4.** SCF-Cdc4 regulates Ame1-Okp1 protein levels in vivo. (**A**) Model of substrate binding to SCF complexes. SCF is composed of Skp1, Cdc53 (Cullin), Rbx1, an F-box protein (e.g., Cdc4 or Grr1), and here with the E2 enzyme Cdc34. Lower panel: overexpression of Ame1-WT leads to accumulation of the protein in a *skp1-3* mutant strain over time as compared to a wild-type background. (**B**) Protein levels of Ame1-Okp1 in different SCF mutants after overexpression. Note that Ame1 levels remain low in the *grr1Δ* mutant (cytoplasmic F-box protein), and that Okp1 strongly

*Figure 4 continued on next page*

*Figure 4 continued*

accumulates in the *cdc4-1* mutant. All alleles were used at the permissive temperature of 30°C. Quantification of Okp1 protein levels for this experiment is shown above, Okp1-WT signal was set to 1. (**C**) Serial dilution assay of overexpressed Ame1-Okp1 variants in wild-type or SCF mutant strain backgrounds (*skp1-3* or *cdc34-2*). Plates were photographed after 2 days at the indicated temperature. (**D**) Serial dilution assay of overexpressed Ame1-Okp1 variants together with Ctf19-Mcm21 in a wild-type strain background. Plates were photographed after 2 days at the indicated temperature.

phosphorylated threonine at position 52 to Cdc4 (sequence VQPIL**TP**PKL, as in the Ser-phosphorylated peptides, three replicas, 100 ns each). The analysis of the resulting trajectories indicated that the peptide establishes conserved interactions at the binding pocket of the protein and remains bound (*Video 4*, *Figure 5—figure supplement 1A*). A representative snapshot of the most populated cluster of structures, along with the overall population of that cluster, is shown in *Figure 5—figure supplement 1B*. The phosphorylated threonine Thr52 establishes a strong hydrogen bond network with the conserved arginine residues of Cdc4: Arg467, Arg485, and Arg534 (*Figure 5A*). Further, Leu56 of the peptide interacts with Arg443 and Thr465 of Cdc4, while the peptide residues Val47, Pro53, and Pro54 also interact with protein residues (*Figure 5—figure supplement 1C*). Additionally, several van der Waals contacts are established between the peptide (through Pro49, Leu51, Lys55, and Ile50) and Cdc4 (involving Leu637, Thr677, Ile676, Trp717, Ser464, Tyr574, and Gly636), indicating an optimal fit of the peptide at the protein binding site. Overall, this strong network of peptide-protein interactions indicates that the VQPIL**TP**PKL peptide is predicted to be a potent Cdc4 binder, even more than the doubly phosphorylated Ser41/Ser45 peptide.

Next, we evaluated the Ame1-CPD$^{ILTPP}$ mutation in the GAL overexpression setting. Strikingly, neither Ame1 nor Okp1 protein was detectable by western blotting upon overexpression under these conditions, and preventing phosphorylation on the five remaining sites (Ame1-5A-CPD$^{ILTPP}$) did not stabilize the protein (*Figure 5B*). If the strong CPD indeed exerts its effect via Cdc4-dependent recognition, then the protein levels of Ame1-CPD$^{ILTPP}$ should be restored to wild-type in an SCF mutant. Combining the Ame1-CPD$^{ILTPP}$ allele with the *cdc4-1* mutant demonstrated that this is indeed the case: the Ame1-CPD$^{ILTPP}$ mutant and also Okp1 were detectable and displayed similar levels as the wild-type proteins in a *cdc4-1* background (*Figure 5B*). These experiments provide evidence that protein levels of Ame1 and Okp1 are regulated by activation of phospho-degrons in an SCF-Cdc4-dependent manner. We also tested the effect of changing motif 1 into a strong CPD in the context of endogenous Ame1. Interestingly, Ame1-CPD$^{ILTPP}$ expressed as the sole copy of Ame1 yielded a viable strain, which showed slight temperature sensitivity at 37°C and increased sensitivity to benomyl (*Figure 5C*). Upon combination with a *cdc4-1* mutant, however, Ame1-CPD$^{ILTPP}$ was able to partially suppress the growth defect of *cdc4-1* at 34°C and improved growth in the presence of hydroxyurea and benomyl. We also examined the relationship between the Ame1-CPD$^{ILTPP}$ allele and other inner kinetochore mutants. Ame1-CPD$^{ILTPP}$ only very modestly aggravated the growth defect of a *ctf19* deletion mutant at low temperatures (20°C). The serial dilution assays also showed that differences in benomyl hypersensitivity between Ame1 wild-type and Ame1-CPD$^{ILTPP}$ became apparent at 15 and 20 µg/ml benomyl and were minor compared to the effect of a *ctf19* deletion (*Figure 5D*). Overall, these experiments show that the Ame1-CPD$^{ILTPP}$ allele slightly reduced fitness in a wild-type strain background, but conferred a growth advantage to *cdc4-1* mutants.

## Phospho-degrons of endogenous Ame1 are activated in a cell cycle-dependent manner

In the experiments described above, cells were challenged with increased levels of Ame1 following expression from a GAL promoter. How does this relate to the regulation of endogenous Ame1? To test this, we constructed Ame1 mutants expressed from their endogenous

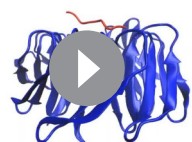

P. Pant & E. Sanchez-Garcia, Computational Biochemistry University of Duisburg-Essen

**Video 4.** Gaussian-accelerated molecular dynamics (GaMD) simulation of the Ame1-CPD$^{ILTPP}$ peptide (red), phosphorylated at T52, binding to the WD40 domain of Cdc4 (blue).
https://elifesciences.org/articles/67390#video4

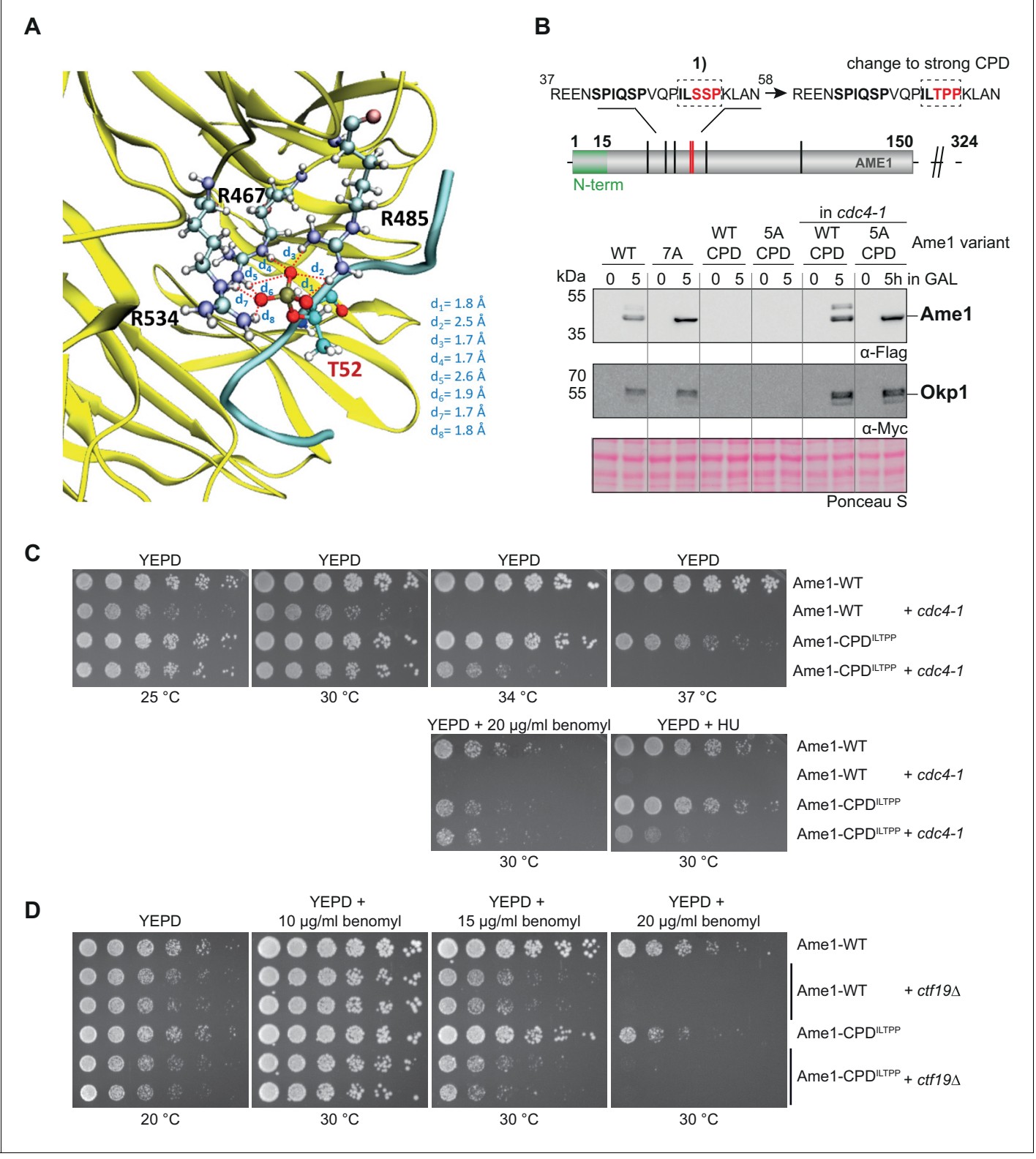

**Figure 5.** Tuning degron strength in the Ame1 N-terminus suppresses a *cdc4* mutant. (**A**) The threonine phosphorylated peptide (VQPILTPPKL, cyan) establishes a strong network of conserved interactions involving its phosphorylated threonine and the conserved arginine residues of Cdc4 (yellow). This binding is further stabilized by several protein-peptide interactions (*Figure 5—figure supplement 1*). (**B**) Changing the phospho-degron motif 1 into a strong Cdc4-degron sequence (ILSSP to ILTPP) leads to a loss of detectable Ame1-CPD in the overexpression system. Also, note that Okp1 is not detectable anymore. The *cdc4-1* mutant background stabilizes Ame1-CPD^ILTPP and Okp1. The *cdc4-1* allele was used at the permissive

*Figure 5 continued on next page*

Figure 5 continued

temperature of 30°C. (C) Serial dilution assay of Ame1-WT or Ame1-CPD$^{ILTPP}$ in a wild-type or *cdc4-1* mutant strain background. Plates were photographed after 3 days of incubation at the indicated temperature. Note that Ame1-CPD partially suppresses the growth defect of *cdc4-1* at 34°C or in the presence of benomyl (20 µg/ml) or hydroxyurea. (D) Serial dilution assays of Ame1-CPD$^{ILTPP}$ combined with a *ctf19* deletion at low temperature (20°C) or increasing benomyl concentrations. Plates were photographed after 2 days (30°C, benomyl) or 3 days (20°C) of incubation at the indicated temperature, respectively. Note the benomyl hypersensitivity of Ame1-CPD$^{ILTPP}$ relative to the wild-type allele.

The online version of this article includes the following figure supplement(s) for figure 5:

**Figure supplement 1.** Analysis of an Ame1 peptide with increased degron strength.

promoter. To simplify the complex phosphorylation pattern of wild-type Ame1 (*Figure 6—figure supplement 1A, C*), we generated alanine mutants that either allowed phosphorylation of the motifs 1 and 2, but prevented phosphorylation of the remaining sites (Ame1-3A, CPD only), or, conversely, prevented motif 1 and 2 phosphorylation, but allowed the remaining sites to be phosphorylated (Ame1-4A, CPD null). Western blotting showed that motif 1/2 phosphorylation of endogenous Ame1 was cell cycle dependent (*Figure 6A*). S-phase-arrested cells displayed a single slowly migrating Ame1 form in addition to unmodified Ame1, while M-phase-arrested cells were maximally phosphorylated with two slowly migrating forms becoming apparent. In the Ame1-4A mutant (CPD null), all slowly migrating forms were eliminated (*Figure 6A*). We followed Ame1 phosphorylation over the course of the cell cycle after release from alpha-factor. Consistent with the analysis of the arrests, we observed that motif 1/2 phosphorylation occurred in a stepwise manner with fully phosphorylated forms appearing 30 min into the cell cycle. FACS analysis demonstrated that by this time cells had completed replication and were in M-phase with a 2C DNA content (*Figure 6B*). Strikingly, phosphorylated Ame1 forms gradually diminished between minute 30 and 60 and only unmodified Ame1 and a single slowly migrating form remained (*Figure 6B*). This indicates that degron motifs on authentic Ame1 are phosphorylated in a cell cycle-dependent manner and lead to disappearance of the phosphorylated forms in M-phase. We additionally analyzed Pds1 degradation kinetics in different Ame1 mutants over the cell cycle. Ame1-3A phosphorylation was highly reproducible, while Ame1-4A eliminated all phosphorylation events observable by shift (*Figure 6C, D*). Interestingly, Ame1 mutants in which only phosphorylation on motif 1 was allowed (Ame1-5A1) or only on motif 2 (Ame1-5A2) also largely eliminated the characteristic stepwise phosphorylation of Ame1-3A (*Figure 6—figure supplement 1D, E*), showing that full phosphorylation is mutually dependent on the presence of both motif 1 and motif 2. Compared to wild-type, neither of the mutants led to a strong delay in the cell cycle as judged by Pds1 degradation kinetics or FACS analysis.

To test the involvement of SCF in the regulation of endogenous Ame1, we took advantage of distinct functionalities of the *skp1-3* allele: At the semipermissive temperature of 34°C, *skp1-3* cells completed replication with nearly wild-type kinetics and accumulated phosphorylated Ame1-3A (*Figure 6E*). In contrast to wild-type cells, however, the phosphorylated Ame1 forms persisted and the cells remained in mitosis with a 2C DNA content. When assayed at the restrictive temperature of 37°C, *skp1-3* cells only slowly progressed to 2C DNA content, indicating they had problems initiating and completing replication. In this situation, Ame1-3A showed only a singly phosphorylated form in western blotting and was only slowly phosphorylated to full extent (*Figure 6F*). Taken together, this experiment indicates that mitotic Skp1 function is required to eliminate endogenous Ame1 phosphorylated at the motifs 1 and 2.

## Binding of the Mtw1 complex shields Ame1 phospho-degrons from Cdk1 phosphorylation

Our analysis described above suggests that Cdk1 phosphorylation targets Ame1 for degradation during mitosis via SCF. This seems counterintuitive, given the critical function of the kinetochore in mitosis. The western blot analysis also indicates, however, that only a subset of endogenous Ame1 is subjected to phosphorylation. Since the phosphorylation cluster is close to the Mtw1 binding site, we asked how binding of the Mtw1c to Ame1 affects phosphorylation by Cdc28-Clb2 in vitro (*Figure 7A*). Analysis of in vitro kinase assays revealed that while Ame1 was heavily phosphorylated in isolation, inclusion of the Mtw1 complex strongly reduced the incorporation of phosphate into Ame1 under identical conditions (*Figure 7B*). This reduction was not due to competing

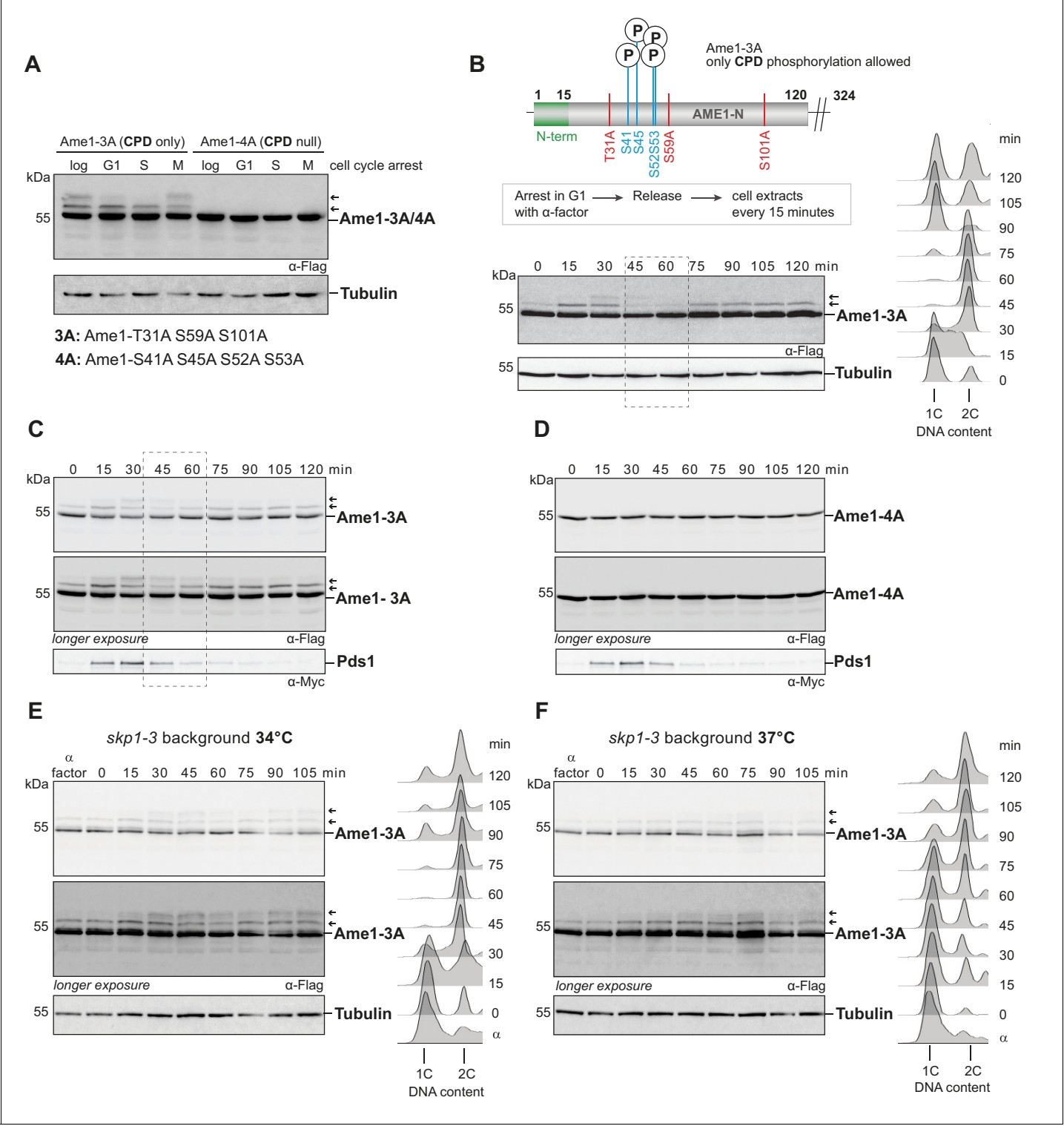

**Figure 6.** Analysis of endogenous Ame1 phospho-mutants over the cell cycle. (**A**) Ame1-variants (Ame1-3A, CPD only, allowing phosphorylation at degron motifs 1 and 2 or eliminating it, Ame1-4A, CPD null) were expressed from the endogenous promoter as the sole copy, and phosphorylation was analyzed in different cell cycle arrests. Drugs used for the arrests: alpha-factor (1 mg/ml) for G1, hydroxyurea (0.2 M) for S-phase, and nocodazole (15 µg/ml) for M-phase. Ame1-3A shows one slowly migrating form in S-phase and two in M-phase, whereas Ame1-4A eliminates all slowly migrating forms. (**B**) Ame1-3A was released from an alpha-factor arrest, and phosphorylation was analyzed by western blotting. Right panel: DNA content analysis by FACS. Phosphorylation is maximal after 30 min when cells have completed S-phase (right: FACS analysis 15 + 30 min), and phosphorylated forms disappear when cells are in mitosis (45 + 60 min, dashed box). For phosphorylation pattern of Ame1-WT, see *Figure 6—figure supplement 1A*. (**C**, **D**)

*Figure 6 continued on next page*

*Figure 6 continued*

Cell cycle analysis of Ame1-3A (C) and Ame1-4A (D). (E) Analysis of Ame1-3A in the *skp1-3* mutant at 34°C (semi-permissive). Note that phosphorylation at motif 1+ 2 persists in the mutant and cells remain in mitosis with 2C DNA content. (F) Analysis of Ame1-3A in the *skp1-3* mutant at 37°C (restrictive). Note that under these conditions cells are delayed to complete replication and mainly a single phospho-form of Ame1 is found.

The online version of this article includes the following figure supplement(s) for figure 6:

**Figure supplement 1.** Cell cycle analysis of Ame1-WT and phosphorylation mutants.

---

phosphorylation of the Mtw1c as only the Dsn1 subunit seemed to be a minor substrate of Cdc28-Clb2. We conclude that free AO is the preferred substrate for Cdk1 phosphorylation.

To analyze the effect of Mtw1c binding on Ame1 phosphorylation in cells, we combined Ame1-3A or Ame1-4A with Mtw1-FRB, which can be anchored away from the nucleus into the cytoplasm upon addition of rapamycin (*Figure 7C*). We observed that upon removal of Mtw1 from the nucleus, phosphorylation at Ame1 degron motifs 1/2 gradually increased over time (*Figure 7D*). FACS analysis indicated that this increase was not due to a major cell cycle effect caused by Mtw1c removal from the nucleus (*Figure 7—figure supplement 1A*). Using this anchor-away setting, we also analyzed the stability of phospho-forms of Ame1 in the presence (-Rapamycin) or absence of the binding partner Mtw1c (+Rapamycin). Upon inhibition of protein synthesis by cycloheximide, slowly migrating forms of Ame1 disappeared over time when Mtw1 was removed from the nucleus (*Figure 7E*). As an independent means to generate unbound Ame1, we also analyzed the phosphorylation status of Ame1 mutants lacking the N-terminal Mtw1c interaction domain. We found that Ame1 variants in which motif1/2 phosphorylation is permitted (Ame1-3A-ΔN) display a prominent slowly migrating form, which is eliminated in the Ame1-4A-ΔN mutant (*Figure 7—figure supplement 1B*). Taken together, these observations are consistent with the notion that unbound Ame1 is the preferred substrate for phosphorylation at motifs 1 and 2 in vivo and that these complexes are susceptible to degradation.

## Discussion

This study reveals novel aspects of phospho-regulation at the budding yeast inner kinetochore. We show that an important function of Cdk1 phosphorylation is to generate phospho-degron motifs on selected inner kinetochore subunits, including the essential COMA subunit Ame1, which are then recognized by the conserved ubiquitin ligase complex SCF with its phospho-adapter Cdc4. We note that Ame1 and Mcm21 peptides were also identified in a large-scale proteomic study geared towards enriching peptides simultaneously regulated by phosphorylation and ubiquitination (*Swaney et al., 2013*). In this context, ubiquitination of COMA subunits was detected for Okp1 (on residue Lys57) and Mcm21 (on residue Lys229).

While Cdk1 has been thought to promote kinetochore assembly in most contexts investigated so far, our study indicates that it can also act as a negative regulator of kinetochore assembly by targeting subunits for ubiquitination and subsequent degradation by the proteasome. This seems counterintuitive at first, given that kinetochores perform their essential role in segregating sister chromatids during mitosis. There are, however, important aspects in which the inner kinetochore subunit Ame1 differs from previously studied SCF substrates: while, for example, the Cdk1 inhibitor Sic1 is fully degraded at the G1-S transition to allow replication initiation, our experiments indicate that only a subset of Ame1 is phosphorylated and subjected to the SCF-dependent pathway. Our biochemical experiments furthermore suggest that the pool of Ame1 regulated by this mechanism corresponds to molecules that are not bound to their binding partner within the kinetochore, the Mtw1c. Such excess Ame1 subunits were also present in our GAL-induced overexpression setting, in which we initially characterized the SCF-dependent regulation of Ame1. While this experiment creates an artificial situation in which the cell is challenged with an increased level of Ame1, this scenario likely also applies to the natural kinetochore assembly process. To ensure effective assembly during S-phase, free kinetochore subcomplexes must be present in excess amounts, otherwise they would become limiting for assembly and prevent the effective formation of a new kinetochore. However, these excess subunits could pose two challenges: first, they could favor ectopic assembly, which would lead to genetic instability. This is most critical for complexes that can effectively nucleate the

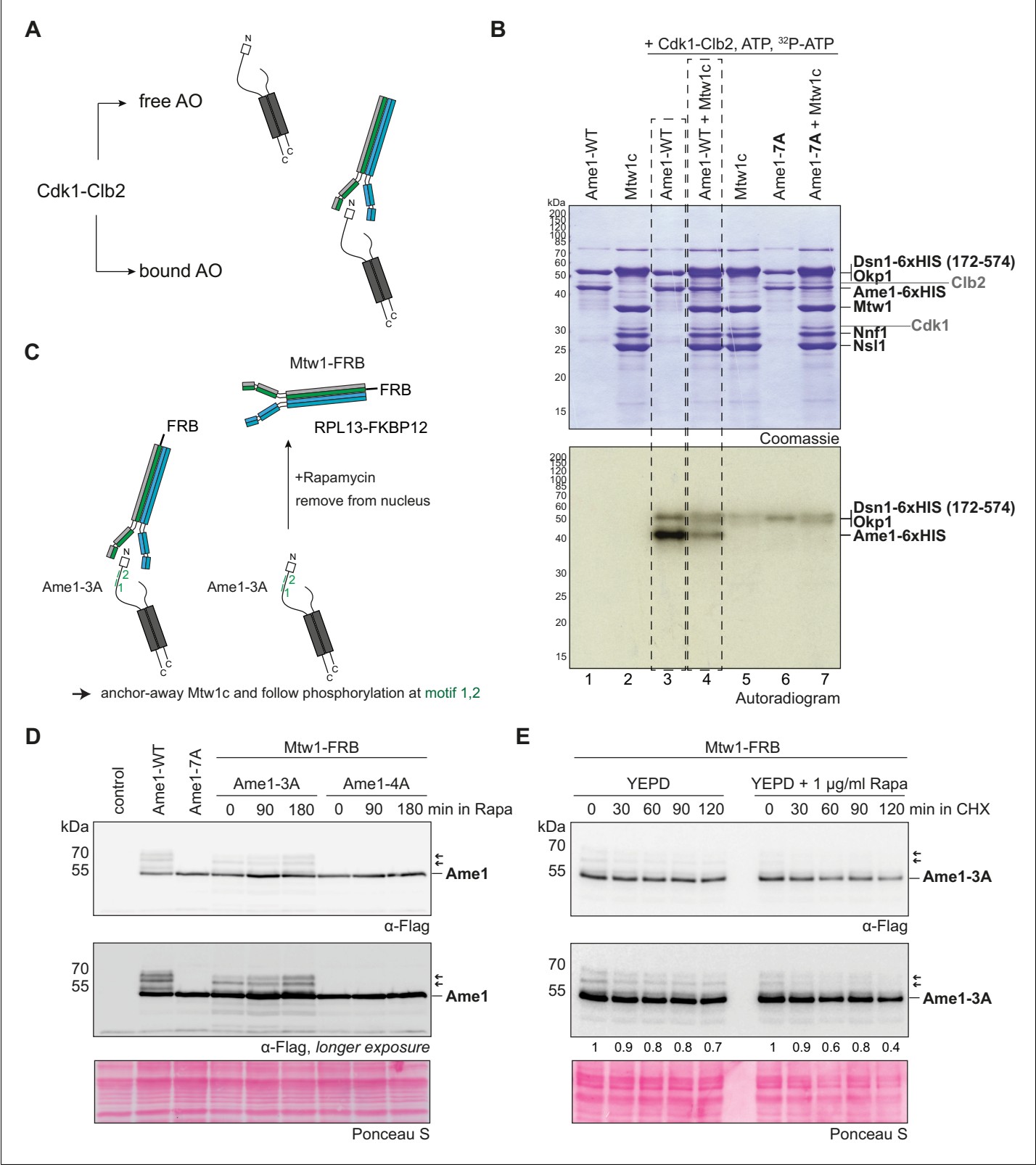

**Figure 7.** Mtw1c binding shields the Ame1 phospho-degron from Cdk1 phosphorylation. (A) Scheme of the kinase assay. Recombinant AO with Ame1-WT or Ame1-7A is used either alone or in combination with its binding partner Mtw1c. (B) In vitro kinase assay of Ame1-Okp1c alone or preincubated with Mtw1c and Cdk1-Clb2 shows decreased phosphorylation of Ame1-WT-Okp1c when bound to Mtw1c (lanes 3 + 4). Phosphorylation of Okp1 is overlapping with phosphorylation of Dsn1 (lanes 4 + 5 + 7). (C) Scheme of the FRB assay. An Mtw1-FRB strain was combined with Ame1-3A or Ame1-4A

*Figure 7 continued on next page*

*Figure 7 continued*

and Ame1 phosphorylation was analyzed after rapamycin addition. Rapamycin anchors Mtw1-FRB out of the nucleus to the ribosomal anchor RPL13-FKBP12. (D) Initial rapamycin assay to follow Ame1 phosphorylation over the time of 0, 90, and 180 min after rapamycin addition. Ame1-3A shows multiple slowly migrating forms that accumulate over time, whereas Ame1-4A eliminates all slowly migrating forms. See *Figure 7—figure supplement 1A* for a corresponding FACS analysis. (E) The rapamycin assay in combination with cycloheximide (CHX). Cultures were preincubated in YEPD for 2 hr and in YEPD + rapamycin for 180 min. New protein translation was inhibited by adding CHX (50 µg/ml) to the medium. Cell extracts were prepared after 0, 30, 60, 90, or 120 min after CHX addition, and Ame1-phosphorylation was analyzed by western blot analysis. Numbers below indicate signal intensities of the slowly migrating forms of Ame1 of the individual timepoints, normalized to timepoint 0.

The online version of this article includes the following figure supplement(s) for figure 7:

**Figure supplement 1.** Additional analysis of Ame1 variants lacking binding to the Mtw1 complex.

formation of new kinetochores, such as Cse4. Also COMA might be an important target for regulation in this regard because it is near the top of the assembly hierarchy, contacts multiple other inner kinetochore subunits (*Yan et al., 2019*), and contains DNA-binding elements. Second, excess free subcomplexes may compete with centromere-bound complexes for binding partners, thereby decelerating kinetochore assembly and making it less effective. It may therefore be beneficial for the cell to 'funnel' kinetochore assembly towards the centromere by preferentially destabilizing unassembled complexes. As shown in *Figure 7*, Mtw1c binding of Ame1-Okp1 subcomplexes shields degron phosphorylation in vitro and Mtw1c removal enhances Ame1 phosphorylation at the discovered motifs in vivo. This Mtw1c binding sensitive phosphorylation could ensure that only free, unused subcomplexes are removed by degradation (*Figure 8A*). From a structural standpoint, kinetochore subcomplexes typically combine relatively short, structured segments (often coiled-coil domains) with large unstructured domains that are the preferred targets of phosphorylation. In this context, phospho-degrons could be ideally suited as assembly sensors for kinetochores since they allow to distinguish excess subunits from properly assembled ones. By placing individually weak degron signals on separate subunits, the cell may allow COMA assembly from AO and CM complexes, while phosphorylation of the assembled COMA then creates stronger composite binding sites for Cdc4 (*Figure 8A*).

In summary, we propose the following model for COMA phospho-regulation by Cdc28$^{Cdk1}$ (*Figure 8B*): initial phosphorylation on Ame1 motifs 1 and 2 starts in S-phase but is not complete before M-phase. This ensures that sufficient free subcomplexes are available for kinetochore assembly. In parallel, the observation that individual degrons on AO or CM are weak permits COMA assembly from its subcomplexes. In M-phase, full degron phosphorylation destabilizes assembled COMA complexes, unless they are bound to the Mtw1 complex, which shields the degrons. The proper timing of phosphorylation is critical in this model. Premature phosphorylation of COMA would target it for destruction too early, likely compromising kinetochore assembly in S-phase. This could be the reason why the Ame1 phospho-sites are only gradually phosphorylated in vivo. Conversely, placing the stronger CPD$^{ILTPP}$ phospho-degron into Ame1 compromised growth in otherwise wild-type cells, but partially rescued the growth phenotypes of *cdc4-1* mutants. This argues that indeed key mitotic SCF substrates reside at the inner kinetochore. Our study adds to the description of molecular links between SCF and centromeres. Notably, a recent study has identified Cse4 as an SCF substrate in budding yeast and shown that Cse4 overexpression aggravates the growth defects of SCF mutants (*Au et al., 2020*). Moreover, assembly and turnover of CBF3 complexes are defective in SCF mutants and both complexes contain Skp1 as a subunit (*Kaplan et al., 1997*). Skp1 mutants also influence the levels of the Mis12 complex at human kinetochores (*Davies and Kaplan, 2010*; *Gascoigne and Cheeseman, 2013*), indicating that SCF-mediated regulation may be a conserved aspect of kinetochore biology.

While our experiments strongly implicate Ame1 as a Cdk1 and SCF substrate, they also show that the Ame1-7A mutant effectively prevents phosphorylation in vivo and in vitro, but does not induce a strong mitotic delay such as the *skp1-3* mutant. We speculate that the Ame1-7A mutant is not sufficient to fully prevent the mitotic defects of SCF, and additional phospho-targets must exist. We note that the Ame1-7A mutation leads to an increased Okp1 level in the *cdc34-2* mutant background, suggesting that it contributes to Okp1 level regulation at least under these specific conditions (see *Figure 4B*). On the other hand, Okp1 itself appears to accumulate even more strongly in the *cdc4-1* mutant than Ame1. This makes it likely that additional, yet undiscovered degron signals

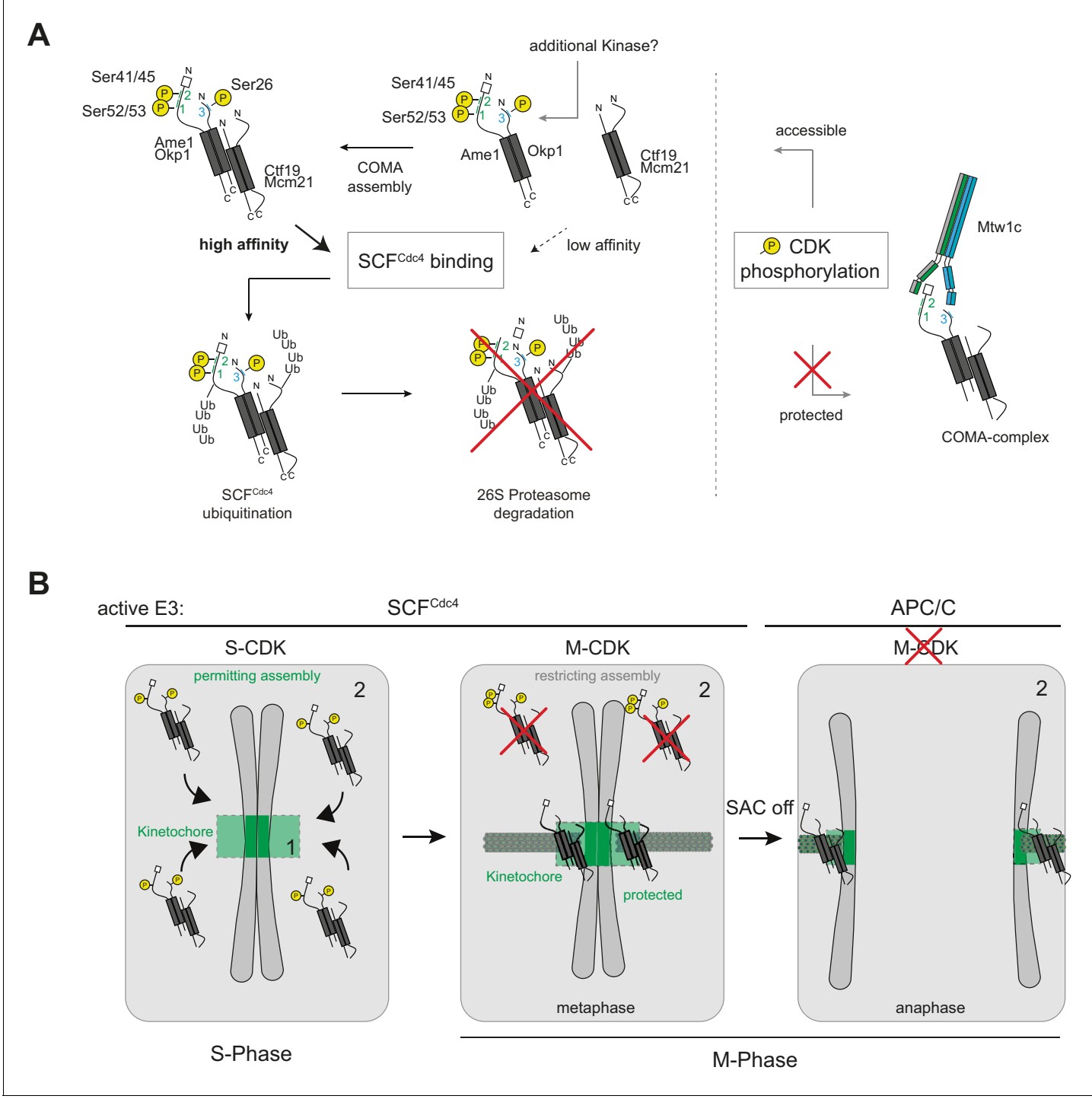

**Figure 8.** Model for SCF-mediated regulation of COMA assembly at the budding yeast kinetochore. (**A**) Scheme illustrating stepwise phosphorylation of the degron motif on Ame1 by Cdk1 and protection from phosphorylation within the kinetochore. In addition to Cdk1, other kinases might be involved in addition, in particular in the phosphorylation of Okp1. (**B**) Cell cycle regulation of COMA complex stability; for details, see discussion. In S-phase, COMA is only partially phosphorylated, allowing assembly at the kinetochore. In M-phase, free COMA is fully phosphorylated and targeted for degradation, while kinetochore-bound COMA is protected.

exist. These could involve both non-consensus sites phosphorylated by Cdk1 and also kinases in addition to Cdk1, as is the case for multiple other SCF substrates (*Faustova et al., 2021*; *Kõivomägi et al., 2011*; *Örd et al., 2019a*). An additional key substrate may be Okp1, and some Cdk1-dependent phosphorylation sites (e.g., Ser70) have been reported (*Holt et al., 2009*), which are not direct Cdk1 sites. Potential kinases include Cdc5/Plk1 and indeed Ame1 residues Ser52/Ser53 constitute candidate Polo-box binding sites, and human CENP-U has been established as a kinetochore receptor for Plk1 (*Singh et al., 2021*). Systematic analysis of kinetochore phosphorylation sites in SCF mutants may help to reveal the most crucial sites in the future. In addition, in vitro reconstitution of kinetochore complex binding to SCF can provide additional information on molecular requirements for binding and ubiquitination.

Previous work has defined the role of SCF-Cdc4 at the G1-S transition, but has also shown that Cdc4, as well as Cdc34 or Cdc53, remain essential genes in *sic1Δ* mutants (*Schwob et al., 1994*), demonstrating that there must be additional key substrates. Furthermore, SCF mutants that complete replication arrest at G2-M with a short spindle and an activated mitotic checkpoint (*Goh and Surana, 1999*; *Schwob et al., 1994*) (our unpublished observations). The relevant substrates for this mitotic function of SCF-Cdc4, however, have so far remained elusive. Our experiments indicate that Ctf19[CCAN] subunits are cell cycle-dependent substrates of SCF-Cdc4 and open up the possibility that regulation of kinetochores by SCF makes a critical contribution to the metaphase-anaphase transition. They also reveal that in addition to the well-established relationship between kinetochores and the E3 ligase APC/C, which regulates the metaphase-anaphase transition via the mitotic checkpoint complex, kinetochores are also molecularly linked to SCF, the other major RING-type E3 ligase complex that regulates the eukaryotic cell cycle.

# Materials and methods

### Key resources table

| Reagent type (species) or resource | Designation | Source or reference | Identifiers | Additional information |
|---|---|---|---|---|
| Strain, strain background (*Saccharomyces cerevisiae*) | S288C | | | |
| Recombinant DNA reagent | See Materials and methods, *Table 2* | | | |
| Genetic reagent, gene (*S. cerevisiae*) | See Materials and methods, *Table 3* | | | |
| Antibody | Anti-Flag (monoclonal, peroxidase conjugated) | Sigma Aldrich | A8592 | 1:10,000 |
| Antibody | Anti-myc 9E10 (mouse monoclonal) | BioLegend | Catalog #626801 | 1:1000 |
| Antibody | Anti-mouse secondary (from sheep) | Cytiva LifeSciences | Catalog #NA931 | 1:10,000 |
| Antibody | Anti-tubulin (monoclonal, peroxidase conjugated) | Santa Cruz Biotechnologies | sc-53030 | 1:1000 |
| Chemical compound, drug | α-factor | | | 1 mg/ml |
| Chemical compound, drug | Hydroxyurea | US Biological Life Sciences | H9120 | 0.2 M |
| Chemical compound, drug | Nocodazole | Sigma Aldrich | M1404-50MG | 15 µg/ml |
| Chemical compound, drug | Benomyl | Sigma Aldrich | 45339-250MG | 15–30 µg/ml |
| Chemical compound, drug | Rapamycin | Diagonal | 370.940.010 | 1 µg/ml |
| Chemical compound, drug | Cycloheximide | Sigma Aldrich | C7698-1G | 50 µg/ml |

## Expression and purification of recombinant proteins (AO, Mtw1c, Cdc28-Clb2)

Expression constructs for kinetochore proteins AO, Mtw1c, and Mtw1-Nnf1 (MN) and the kinase complex Cdc28-Clb2 used in this study were created by amplification of the DNA for the respective genes from yeast genomic DNA and cloning into pETDuet-1, pET3aTr/pST39, pST44 plasmids (bacterial expression) following the protocol for restriction-free cloning or pESC two-micron plasmids (yeast expression) using classical cloning methods. Restriction-free cloning was also used to produce vectors encoding phospho-eliminating mutants in Ame1. Site-directed mutagenesis (Agilent Technologies) was applied for introduction of amino acid substitutions. A list of all vectors used for protein production and purification in bacteria or yeast cells can be found in *Table 3*.

The following conditions were used for protein production and purification unless indicated otherwise: Competent bacterial cells were grown at 37°C until $OD_{600}$ of 0.6 and subsequently induced with 0.5 mM IPTG. Expressions were conducted overnight at 18–20 °C. Expression of Mtw1c and MN constructs was performed in BL21 (DE3; Novagen) and AO was expressed in Rosetta 2(pLys) (DE3) cells (EMD Millipore). Cdc28-Clb2 was overexpressed in the protease-deficient yeast strain DDY1810 overnight at 30°C using 2% galactose. Lysis, wash, and elution buffers as well as chromatography steps varied for the different protein complexes and are described in the individual sections. The Poly-Histidine fusion proteins were isolated with HisTrap HP 5 ml columns. The kinase complex was isolated using IgG Sepharose 6 Fast Flow and calmodulin sepharose 4B affinity resin (both GE Healthcare). In the last step of all bacterially expressed proteins, the protein was passed over a size-exclusion chromatography (SEC) column that was appropriate for the proteins size while elution of proteins was measured by absorbance at 280 nm using an Äkta FPLC System connected to a Windows-based laptop running the UNICORN control software. All proteins were concentrated to the desired concentration except the kinase complex and flash frozen in liquid nitrogen before stored at −80°C until usage.

## Mtw1c

The Mtw1 complex (Mtw1c) used in this study includes an N-terminal 171 amino acid truncation of the Dsn1 subunit and an N-terminal 6xHis tag at the same protein. Lysis buffer for Mtw1c was 50 mM Tris-HCl, pH 7.5, 500 mM NaCl, 30 mM imidazole, 10% glycerol, and 5 mM β-mercaptoethanol as described previously (*Maskell et al., 2010*). Lysates were loaded onto a HisTrap HP 5 ml column pre-equilibrated in lysis buffer, washed with 30 column volumes (CV) lysis buffer, and eluted with 30 mM Tris, pH 8.5, 80 mM NaCl, 10% glycerol, 5 mM β-mercaptoethanol, and 250 mM imidazole. Subsequently, proteins were directly loaded onto anion exchange chromatography (HiTrap Q HP 5 ml column; GE Healthcare). The column was equilibrated with 4 CV gradient wash from 0–20% buffer B, followed by 15 CV wash of 20% buffer B. The chromatography was performed with a gradient consisting of buffer (30 mM Tris HCl, pH 8.5, and 5% glycerol from 80 mM [A] to 1 M NaCl [B]) using a flow rate of 1 ml/min and an elution volume of 40 CV. The protein is eluted at a buffer B ratio of about 40%. Fractions containing the Mtw1c were concentrated using a Vivaspin 20 ultracentrifugal unit MWCO 10000 Dalton (Satorius) and loaded to a HiLoad Superdex 200 16/600 pg (GE Healthcare) equilibrated in 30 mM HEPES, pH 7.5, 250 mM NaCl, 5% glycerol, and 2 mM TCEP.

## AO

Bacterial pellets expressing the Ame1-Okp1 complex (AO) were resuspended in lysis buffer (30 mM HEPES, pH 7.5, 30 mM imidazole [A] or 1 M imidazole [B], 600 mM NaCl, and 5 mM β-mercaptoethanol). AO loaded on the HisTrap HP 5 ml column was washed with 30 CV lysis buffer before eluted with 300 mM imidazole in the otherwise same buffer or using an imidazole gradient from 0% to 60%. AO eluted between 150 mM and 300 mM imidazole. Subsequently, the complex was purified over gel filtration Superdex 200 10/300 GL or HiLoad Superdex 200 16/600 pg columns (GE Healthcare) in 30 mM HEPES, pH 7.5, 250 mM NaCl, 5% glycerol, and 2 mM TCEP. The complex eluted in two peaks from the column, of which the later eluting peak was used for subsequent experiments. All used amino acid substitutions were purified in the same way as described above.

**Table 2.** Vectors for protein expression and yeast strain generation.

| Plasmid | Description | Source |
|---|---|---|
| *Bacterial expression* | | |
| pSW698 | pST39-Mtw1/Nsl1/Nnf1/6xHis-Dsn1 | *Hornung et al., 2014* |
| pSW900 | pST39-Okp1/Ame1-6xHis | *Hornung et al., 2014* |
| pMLU16 | pST39-Okp1/Ame1-7A-6xHis (T31A, S41A, S45A, S52A, S53A, S59A, S101A) | This study |
| pMB91 | pST39-Okp1/Ame1-S52A+S53A-6xHis | This study |
| pMB92 | pST39-Okp1/Ame1-T31A-6xHis | This study |
| pMB93 | pST39-Okp1/Ame1-S41A+S45A-6xHis | This study |
| pMB94 | pST39-Okp1/Ame1-4A-6xHis (S41A, S45A, S52A, S53A) | This study |
| pMB109 | pST39-Okp1-S26A/Ame1-6xHis | This study |
| pMB110 | pST39-Okp1-S26A/Ame1-4A-6xHis (S41A, S45A, S52A. S53A) | This study |
| pMB111 | pST39-Okp1-S26A/Ame1-7A-6xHis (T31A, S41A, S45A, S52A, S53A, S59A, S101A) | This study |
| pMB112 | pST39-Okp1-WT/Ame1-3A-6xHis (T31A, S59A, S101A) | This study |
| *Yeast expression* | | |
| | pESC-Clb2-TAP | Morgan Lab |
| | pESC-Clb5-TAP | Morgan Lab |
| *Yeast genetics* | | |
| pMLU13 | Ame1-7A-6xFlag in pRS306 (T31A, S41A, S45A, S52A, S53A, S59A, S101A) | This study |
| pMLU17 | Ame1-7E-6xFlag in pRS306 (T31E, S41E, S45E, S52E, S53E, S59E. S101E) | This study |
| pMB54 | Ame1-WT-1xFlag + Okp1-WT-1xMyc in pESC-HIS | This study |
| pMB55 | Ame1-7A-1xFlag + Okp1-WT-1xMyc in pESC-HIS | This study |
| pMB56 | Ame1-7E-1xFlag + Okp1-WT-1xMyc in pESC-HIS | This study |
| pMB64 | Ame1-TM3-6xFlag in pRS306 (Δ31–89) | This study |
| pMB65 | Ame1-TM4-6xFlag in pRS306 (Δ31–116) | This study |
| pMB66 | Ame1-TM5-6xFlag in pRS306 (Δ31–187) | This study |
| pMB68 | Ame1-TM2-6xFlag in pRS306 (Δ31–75) | This study |
| pMB72 | Ame1-S41A+S45A-1xFlag + Okp1-WT-1xMyc in pESC-HIS | This study |
| pMB73 | Ame1-S52A+S53A-1xFlag + Okp1-WT-1xMyc in pESC-HIS | This study |
| pMB84 | Ame1-ILTPP-1xFlag + Okp1-WT-1xMyc (optimal CPD, S52T + S53P) in pESC-HIS | This study |
| pMB85 | Ame1-5A+ILTPP-1xFlag + Okp1-WT-1xMyc (optimal CPD, T31A, S41A, S45A, S52T, S53P, S59A, S101A)in pESC-HIS | This study |
| pMB86 | Ame1-T31A-1xFlag + Okp1-WT-1xMyc in pESC-HIS | This study |
| pMB87 | Ame1-S45A-1xFlag + Okp1-WT-1xMyc in pESC-HIS | This study |
| pMB90 | Ame1-S41A-1xFlag + Okp1-WT-1xMyc in pESC-HIS | This study |
| pMB98 | pRS306-Ame1-3A-6xFlag (T31A, S59A, S101A) | This study |
| pMB99 | pRS306-Ame1-4A-6xFlag (S41A, S45A, S52A, S53A) | This study |
| pMB104 | Ame1-7A-1xFlag + Okp1-S26A-1xMyc in pESC-HIS | This study |
| pMB113 | pRS306-Ame1-5A1-6xFlag (T31A, S41A, S45A, S59A, S101A) | This study |
| pMB114 | pRS306-Ame1-5A2-6xFlag (T31A, S52A, S53A, S59A, S101A) | This study |
| pMB156 | pRS306-Ame1-3A-ΔN-6xFlag (Δ2–15, T31A, S59A, S101A) | This study |
| pMB157 | pRS306-Ame1-4A-ΔN-6xFlag (Δ2–15, S41A, S45A, S52A. S53A) | This study |

*Table 2 continued on next page*

*Table 2 continued*

| Plasmid | Description | Source |
|---------|-------------|--------|
| pSW731 | pRS306-Ame1-WT-6xFlag | *Hornung et al., 2014* |
| pESC-HIS | pESC-HIS | Agilent Technologies |
| pESC-URA | pESC-URA | Agilent Technologies |
| pDD526 | Pds1-13xMyc in pRS305 | Westermann lab |

## Cdc28-Clb2

Yeast cells were incubated in minimal medium under constant selective pressure (SD or S-RG doURA). Expression was induced by adding 2% galactose to S-R doURA medium containing 2% raffinose and additionally 10× YEP solution to boost cell growth. Yeast pellets were flash frozen in liquid nitrogen as droplets and grind into powder using a freezer mill (SPEX SamplePrep 6870). Powder was lysed in 1× Hyman buffer (50 mM bis-Tris propane-HCl pH 7.0, 100 mM KCl, 5 mM EDTA, 5 mM EGTA, 10% glycerol; additional added protease inhibitors; Pierce Thermo Fisher, Calbiochem and 1 mM PMSF) with 1% Triton X-100 using sonication. The lysate after high-speed centrifugation was incubated with pre-equilibrated IgG Sepharose 6 Fast Flow affinity resin (washed in TST [50 mM Tris-HCl pH 7.4, 150 mM NaCl, 0.1% Tween 20] and $NH_4OAc$ pH 3.4) for 4 hr at 4°C and the salt concentration was increased to 300 mM KCl. Beads were washed with 50 bead volumes 1× Hyman buffer and proteins were eluted with TEV-protease overnight at 4°C. TEV-Eluate (with increase $CaCl_2$) was further incubated using pre-equilibrated calmodulin sepharose 4B affinity resin in calmodulin binding buffer (25 mM Tris-HCl pH 8.0, 150 mM NaCl, 0.02% NP-40, 1 mM $MgCl_2$, 1 mM imidazole, 2 mM $CaCl_2$, 1 mM DTT) for 1–2 hr at 4°C. Beads were washed in 40 bead volumes calmodulin binding buffer and eluted in small fractions (300 µl) using calmodulin elution buffer (like calmodulin binding buffer, 2 mM $CaCl_2$ is substituted with 20 mM EGTA). Single fractions were flash frozen in liquid nitrogen and stored at −80°C until further usage in in vitro kinase assays.

## Yeast genetics (strain construction, FRB system, pESCs, integration/replacement constructs)

Yeast strains were constructed in the S288C or W303 (SCF strains) background. A list of all yeast strains used in this study can be found in *Table 3*, a list of all vectors used for generation of novel yeast strains can be found in *Table 2*. Yeast strain generation and methods were performed by standard procedures.

The anchor-away approach for characterization of Ame1 in SCF mutants or wild-type strain was performed as described (*Haruki et al., 2008*) using the ribosomal RPL13-FKBP12 anchor. Final rapamycin concentration in plates or liquid media was 1 µg/ml. For protein stability assays, cells were treated with rapamycin for 180 min, followed by the addition of cycloheximide (CHX) with a final concentration of 50 µg/ml. Serial twofold dilutions of overnight cultures were prepared on 96-well plates in minimal medium starting from $OD_{600}$ of 0.4 for anchor-away approach or 0.5 for overexpression approach. The dilutions were spotted on YPD medium with and without rapamycin or on minimal medium with either glucose or raffinose + galactose (2% each) and grown at 30°C for 2–3 days. To confirm phenotypes observed in the serial dilution assays for Ame1 mutants, Ame1 hemizygous deletion strains were used to introduce Ame1 wild-type or phospho-mutants at an exogenous locus before haploid spores were produced. Pds1-13xMyc was integrated exogenously into haploid strains for cell cycle experiments. For overexpression of proteins, pESC two-micron plasmids were transformed into haploid wild-type or SCF mutant strains without integration into the genome and clone pools were used for further analyses. Selective pressure was used for maintenance of the plasmids. Expression of integrated proteins was checked for all created yeast strains by protein extraction from yeast (*Kushnirov, 2000*) and western blotting against the respective tags of individual proteins (see Key resources table *Table 2*).

**Table 3.** Yeast strains.

| Strain name | Relevant genotype | Source |
|---|---|---|
| SWY355 | Mat a; tor1-1, fpr1::loxP-LEU2-loxP, RPL13A-2xFKBP12::loxP-TRP1-loxP | *Haruki et al., 2008* |
| SWY536 | Mat a; tor1-1 fpr1::loxP-LEU2-loxP RPL13A-2xFKBP12::loxP, Ame1-FRB::KanMX | *Hornung et al., 2014* |
| PSY1.1 | Mat a; tor1-1 fpr1::loxP-LEU2-loxP RPL13A-2xFKBP12::loxP, Mtw1-FRB::KanMX | *Killinger et al., 2020* |
| MLY3 | Mat a; tor1-1 fpr1::loxP-LEU2-loxP RPL13A-2xFKBP12::loxP, Ame1-FRB::KanMX, Ame1-WT-6xFlag::URA3 | This study |
| MLY5 | Mat a; tor1-1 fpr1::loxP-LEU2-loxP RPL13A-2xFKBP12::loxP, Ame1-FRB::KanMX, Ame1-7A-6xFlag::URA3 | This study |
| MLY15 | Mat a, ade2-1, leu2-3,112, his3Δ200, ame1Δ::HIS3, ura3-52, Ame1-WT-6xFlag::URA3 | This study |
| MLY31 | Mat a; tor1-1 fpr1::loxP-LEU2-loxP RPL13A-2xFKBP12::loxP, Ame1-FRB::KanMX, Ame1-7E-6xFlag::URA3 | This study |
| MBY79 | Mat a, ade2-1, his3Δ200, ame1Δ::HIS3, leu2-3,112, ura3-52, Ame1-WT-6xFlag::URA3, Cse4-13xMyc::KanMX | This study |
| MBY81 | Mat a, ade2-1, his3Δ200, ame1Δ::HIS3, leu2-3,112, ura3-52, Ame1-7E-6xFlag::URA3, Cse4-13xMyc::KanMX | This study |
| MBY83 | Mat a, ade2-1, his3Δ200, ame1Δ::HIS3, leu2-3,112, ura3-52, Ame1-7A-6xFlag::URA3, Cse4-13xMyc::KanMX | This study |
| MBY153 | Mat α, lys2-801am, leu2-3,112, his3Δ200, ura3-52 (two-micron-pESC-HIS) (pGAL-empty) | This study |
| MBY155 | Mat α, lys2-801am, leu2-3,112, his3Δ200, ura3-52 (two-micron pESC-HIS-Ame1-WT-1xFlag + Okp1-WT-1xMyc) | This study |
| MBY156 | Mat α, lys2-801am, leu2-3,112, his3Δ200, ura3-52 (two-micron pESC-HIS-Ame1-7A-1xFlag + Okp1-WT-1xMyc) | This study |
| MBY157 | Mat α, lys2-801am, leu2-3,112, his3Δ200, ura3-52 (two-micron pESC-HIS-Ame1-7E-1xFlag + Okp1-WT-1xMyc) | This study |
| MBY158 | Mat α, lys2-801am, leu2-3,112, his3Δ200, ura3-52, psh1Δ::NatNT2 (two-micron pESC-HIS) | This study |
| MBY160 | Mat α, lys2-801am, leu2-3,112, his3Δ200, ura3-52, psh1Δ::NatNT2 (two-micron pESC-HIS-Ame1-WT-1xFlag + Okp1-WT-1xMyc) | This study |
| MBY161 | Mat α, lys2-801am, leu2-3,112, his3Δ200, ura3-52, psh1Δ::NatNT2 (two-micron pESC-HIS-Ame1-7A-1xFlag + Okp1-WT-1xMyc) | This study |
| MBY162 | Mat α, lys2-801am, leu2-3,112, his3Δ200, ura3-52, psh1Δ::NatNT2 (two-micron pESC-HIS-Ame1-7E-1xFlag + Okp1-WT-1xMyc) | This study |
| MBY163 | Mat α, lys2-801am, leu2-3,112, his3Δ200, ura3-52, mad1Δ::KanMX (two-micron pESC-HIS) | This study |
| MBY165 | Mat α, lys2-801am, leu2-3,112, his3Δ200, ura3-52, mad1Δ::KanMX (two-micron pESC-HIS-Ame1-WT-1xFlag + Okp1-WT-1xMyc) | This study |
| MBY166 | Mat α, lys2-801am, leu2-3,112, his3Δ200, ura3-52, mad1Δ::KanMX (two-micron pESC-HIS-Ame1-7A-1xFlag + Okp1-WT-1xMyc) | This study |
| MBY167 | Mat α, lys2-801am, leu2-3,112, his3Δ200, ura3-52, mad1Δ::KanMX (two-micron pESC-HIS-Ame1-7E-1xFlag + Okp1-WT-1xMyc) | This study |
| MBY225 | Mat α, his3Δ200, ura3-52, Ame1-TM2-6xFlag::URA3, lys2-801am, tor1-1 fpr1::loxP-LEU2-loxP RPL13A-2xFKBP12::loxP-TRP1-loxP, Ame1-FRB::KanMX | This study |
| MBY226 | Mat α, his3Δ200, ura3-52, Ame1-TM3-6xFlag::URA3, lys2-801am, tor1-1 fpr1::loxP-LEU2-loxP RPL13A-2xFKBP12::loxP-TRP1-loxP, Ame1-FRB::KanMX | This study |
| MBY227 | Mat α, his3Δ200, ura3-52, Ame1-TM4-6xFlag::URA3, lys2-801am, tor1-1 fpr1::loxP-LEU2-loxP RPL13A-2xFKBP12::loxP-TRP1-loxP, Ame1-FRB::KanMX | This study |
| MBY228 | Mat α, his3Δ200, ura3-52, Ame1-TM5-6xFlag::URA3, lys2-801am, tor1-1 fpr1::loxP-LEU2-loxP RPL13A-2xFKBP12::loxP-TRP1-loxP, Ame1-FRB::KanMX | This study |
| MBY241 | Mat α, lys2-801am, leu2-3,112, his3Δ200, ura3-52 (two-micron pESC-HIS-Ame1-S52A+S53A-1xFlag + Okp1-WT-1xMyc) | This study |
| MBY255 | Mat α, lys2-801am, leu2-3,112, his3Δ200, ura3-52 (two-micron pESC-HIS-Ame1-ILTPP-1xFlag + Okp1-WT-1xMyc) | This study |
| MBY256 | Mat α, lys2-801am, leu2-3,112, his3Δ200, ura3-52 (two-micron pESC-HIS-Ame1-5A+ILTPP-1xFlag + Okp1-WT-1xMyc) | This study |
| MBY273 | Mat a, ade2-1, leu2-3,112, his3Δ200, ame1Δ::HIS3, ura3-52, Ame1-WT-CPD$^{ILTPP}$-6xFlag::URA3 | This study |
| MBY275 | Mat α, lys2-801am, leu2-3,112, his3Δ200, ura3-52 (two-micron pESC-HIS-Ame1-T31A-1xFlag + Okp1-WT-1xMyc) | This study |
| MBY276 | Mat α, lys2-801am, leu2-3,112, his3Δ200, ura3-52 (two-micron pESC-HIS-Ame1-S45A-1xFlag + Okp1-WT-1xMyc) | This study |
| MBY278 | Mat α, lys2-801am, leu2-3,112, his3Δ200, ura3-52 (two-micron pESC-HIS-Ame1-S41A+S45A-1xFlag + Okp1-WT-1xMyc) | This study |
| MBY279 | Mat α, lys2-801am, leu2-3,112, his3Δ200, ura3-52 (two-micron pESC-HIS-Ame1-S41A-1xFlag + Okp1-WT-1xMyc) | This study |
| MBY292 | Mat a, ura3-52, lys2-801, ade2-101, his3Δ200, trp1Δ63, leu2Δ1, skp1-3::LEU2 (two-micron pESC-HIS-Ame1-WT-1xFlag-Okp1-WT-1xMyc) | This study |

*Table 3 continued on next page*

*Table 3 continued*

| Strain name | Relevant genotype | Source |
|---|---|---|
| MBY293 | Mat a, ura3-52, lys2-801, ade2-101, his3Δ200, trp1Δ63, leu2Δ1, skp1-3::LEU2 (two-micron pESC-HIS-Ame1-7A-1xFlag-Okp1-WT-1xMyc) | This study |
| MBY310 | Mat a, cdc53-1 (ts), ade2-1, trp1-1, can1-100, leu2-3,112, his3-11,15, ura3, psi+, ssd1-d2 (two-micron pESC-HIS-Ame1-WT-1xFlag-Okp1-WT-1xMyc) | This study |
| MBY311 | Mat a, cdc53-1 (ts), ade2-1, trp1-1, can1-100, leu2-3,112, his3-11,15, ura3, psi+, ssd1-d2 (two-micron pESC-HIS-Ame1-7A-1xFlag-Okp1-WT-1xMyc) | This study |
| MBY312 | Mat a, cdc34-2 (ts), ade2-1, trp1-1, can1-100, leu2-3,112, his3-11,15, ura3, psi+, ssd1-d2 (two-micron pESC-HIS-Ame1-WT-1xFlag-Okp1-WT-1xMyc) | This study |
| MBY313 | Mat a, cdc34-2 (ts), ade2-1, trp1-1, can1-100, leu2-3,112, his3-11,15, ura3, psi+, ssd1-d2 (two-micron pESC-HIS-Ame1-7A-1xFlag-Okp1-WT-1xMyc) | This study |
| MBY314 | Mat α, cdc4-1 (ts), ade2-1, trp1-1, can1-100, leu2-3,112, his3-11,15, ura3, psi+, ssd1-d2 (two-micron pESC-HIS-Ame1-WT-1xFlag-Okp1-WT-1xMyc) | This study |
| MBY315 | Mat alpha, cdc4-1 (ts), ade2-1, trp1-1, can1-100, leu2-3,112, his3-11,15, ura3, psi+, ssd1-d2 (two-micron pESC-HIS-Ame1-7A-1xFlag-Okp1-WT-1xMyc) | This study |
| MBY316 | Mata, ade2-1, his3Δ200, trp1-1, ura3-52, grr1Δ::LEU2 (ts), lys2-801 (two-micron pESC-HIS-Ame1-WT-1xFlag-Okp1-WT-1xMyc) | This study |
| MBY317 | Mata, ade2-1, his3Δ200, trp1-1, ura3-52, grr1Δ::LEU2 (ts), lys2-801 (two-micron pESC-HIS-Ame1-7A-1xFlag-Okp1-WT-1xMyc) | This study |
| MBY322 | Mat a, ade2-1, leu2-3,112, his3Δ200, ura3-52, mub1Δ::natNT2 (two-micron pESC-HIS-Ame1-WT-1xFlag-Okp1-WT-1xMyc) | This study |
| MBY323 | Mat a, ade2-1, leu2-3,112, his3Δ200, ura3-52, mub1Δ::natNT2 (two-micron pESC-HIS-Ame1-7A-1xFlag-Okp1-WT-1xMyc) | This study |
| MBY324 | Mat a, his3Δ200, ame1Δ::HIS3, ura3-52, Ame1-3A-6xFlag::URA3, leu2-3-112 | This study |
| MBY327 | Mat a, his3Δ200, ame1Δ::HIS3, ura3-52, Ame1-4A-6xFlag::URA3, leu2-3-112 | This study |
| MBY331 | Mat α, lys2-801am, leu2-3,112, his3Δ200, ura3-52 (two-micron pESC-HIS-Ame1-7A-1xFlag+Okp1-1A-1xMyc) | This study |
| MBY333 | Mat a, ura3-52, lys2-801, ade2-101, his3Δ200, trp1Δ63, leu2Δ1, skp1-3::LEU2 (two-micron pESC-HIS-Ame1-7A-1xFlag+Okp1-1A-1xMyc) | This study |
| MBY345 | Mat a, ame1Δ::HIS3, ura3-52, Ame1-5A1::URA3, ade2-3, leu2-3,112 | This study |
| MBY347 | Mat a, ame1Δ::HIS3, ura3-52, Ame1-5A2::URA3, ade2-3, leu2-3,112 | This study |
| MBY360 | Mat α, cdc4-1, ade2-1, trp1-1, can1-100, leu2-3,112, his3-11,15, ura3, psi+, ssd1-d2 (two-micron pESC-HIS-Ame1-CPD$^{ILTPP}$-1xFlag+Okp1-WT-1xMyc) | This study |
| MBY361 | Mat α, cdc4-1, ade2-1, trp1-1, can1-100, leu2-3,112, his3-11,15, ura3, psi+, ssd1-d2 (two-micron pESC-HIS-Ame1-5A-CPD$^{ILTPP}$-1xFlag+Okp1-WT-1xMyc) | This study |
| MBY371 | Mat a, his3Δ200, ame1Δ::HIS3, ura3-52, Ame1-3A-6xFlag::URA3, leu2-3,112, Pds1-13xMyc::LEU2 | This study |
| MBY372 | Mat α, his3Δ200, ame1Δ::HIS3, ura3-52, Ame1-4A-6xFlag::URA3, leu2-3,112, Pds1-13xMyc::LEU2 | This study |
| MBY373 | Mat a, his3Δ200, ame1Δ::HIS3, ura3-52, Ame1-5A1::URA3, ade2-3, leu2-3,112, Pds1-13xMyc::LEU2 | This study |
| MBY374 | Mat a, his3Δ200, ame1Δ::HIS3, ura3-52, Ame1-5A2::URA3, ade2-3, leu2-3,112, Pds1-13xMyc::LEU2 | This study |
| MBY380 | Mat a, his3Δ200, ame1Δ::HIS3, ura3-52, Ame1-3A-6xFlag::URA3, lys2-801, trp1-1, leu2-3,112, skp1-3::LEU2 | This study |
| MBY412 | Mat α, cdc34-2, ade2-1, trp1-1, can1-100, leu2-3,112, his3-11,15, ura3, psi+, ssd1-d2 (two-micron pESC-HIS-Ame1-7A-1xFlag+Okp1-1A-1xMyc) | This study |
| MBY434 | Mat a, ade2-1, leu2-3,112, his3-11-15, ame1Δ::HIS3, ura3-52, Ame1-WT-CPD$^{ILTPP}$-6xFlag::URA3, psi1x, ssd1-d2, cdc4-1 | This study |
| MBY456 | Mat α, lys2-801am, leu2-3,112, his3Δ200, ura3-52 (two-micron pESC-HIS, two-micron pESC-URA3) | This study |
| MBY457 | Mat α, lys2-801am, leu2-3,112, his3Δ200, ura3-52 (two-micron pESC-HIS, two-micron pGAL-URA-Mcm21-WT-3xHA+Ctf19-WT-1xMyc) | This study |
| MBY462 | Mat α, lys2-801am, leu2-3,112, his3Δ200, ura3-52 (two-micron pESC-HIS-Ame1-WT-1xFlag+Okp1-WT-1xMyc, two-micron pESC-URA) | This study |
| MBY463 | Mat α, lys2-801am, leu2-3,112, his3Δ200, ura3-52 (two-micron pESC-HIS-Ame1-WT-1xFlag+Okp1-WT-1xMyc, two-micron pESC-URA-Mcm21-WT-3xHA+Ctf19-WT-1xMyc) | This study |
| MBY466 | Mat α, lys2-801am, leu2-3,112, his3Δ200, ura3-52 (two-micron pESC-HIS-Ame1-7A-1xFlag+Okp1-1A-1xMyc, two-micron pESC-URA) | This study |
| MBY485 | Mat α, his3Δ200, ura3-52, Ame1-3A-6xFlag::URA3, lys2-801am, tor1-1 fpr1::loxP-LEU2-loxP RPL13A-2xFKBP12::loxP-TRP1-loxP, Mtw1-FRB::KanMX | This study |

*Table 3 continued on next page*

*Table 3 continued*

| Strain name | Relevant genotype | Source |
|---|---|---|
| MBY486 | Mat α, his3Δ200, ura3-52, Ame1-4A-6xFlag::URA3, lys2-801am, tor1-1 fpr1::loxP-LEU2-loxP RPL13A-2xFKBP12::loxP-TRP1-loxP, Mtw1-FRB::KanMX | This study |
| MBY489 | Mat a, ade2-1, leu2-3,112, his3Δ200, ura3-52, Ame1-3A-ΔN-6xFlag::URA3 | This study |
| MBY490 | Mat a, ade2-1, leu2-3,112, his3Δ200, ura3-52, Ame1-4A-ΔN-6xFlag::URA3 | This study |
| MBY493 | Mat a, ade2-101, his3Δ200, trp1Δ63, lys2-801am, leu2Δ1, skp1-3::LEU2, ura3-52, Ame1-3A-ΔN-6xFlag::URA3 | This study |
| MBY494 | Mat a, ade2-101, his3Δ200, trp1Δ63, lys2-801am, leu2Δ1, skp1-3::LEU2, ura3-52, Ame1-4A-ΔN-6xFlag::URA3 | This study |
| MBY505.1 | Mat a, his3Δ200, ame1Δ::HIS3, ura3-52, Ame1-WT-6xFlag::URA3, ade2-1, lys2-801am, leu2-3,112, ctf19Δ::natNT2 | This study |
| MBY505.2 | Mat a, his3Δ200, ame1Δ::HIS3, ura3-52, Ame1-WT-6xFlag::URA3, ade2-1, leu2-3,112, ctf19Δ::natNT2 | This study |
| MBY509.2 | Mat a, his3Δ200, ame1Δ::HIS3, ura3-52, Ame1-CPD$^{ILTPP}$-6xFlag::URA3, leu2-3,112, ctf19Δ::natNT2 | This study |
| MBY510.1 | Mat α, his3Δ200, ame1Δ::HIS3, ura3-52, Ame1-CPD$^{ILTPP}$-6xFlag::URA3, ade2-1, lys2-801am, leu2-3,112, ctf19Δ::natNT2 | This study |

## GAL overexpression

For overexpression studies, strains containing two-micron plasmids were grown overnight in YEPD. Next morning, cells were washed twice in YEP-raffinose (2%) and incubated in YEP + R for 3 hr at 30°C. Overexpression was induced with the addition of 2% galactose and timepoints were taken after 0, 3, and 5 hr in YEP + RG. Protein extracts (*Kushnirov, 2000*) for western blotting analysis were prepared for each timepoint, and accumulation of protein over time was visualized using anti-Flag M2 antibody for Ame1 or anti c-myc (9E10) for Okp1.

## Cell cycle analysis (synchronization, FACS)

For cell cycle experiments, cells were diluted in 80 ml of liquid YEPD with a starting $OD_{600}$ = 0.2 and incubated for 1 hr at 30°C. Next, cells were arrested in a G1-like state with alpha-factor for 2 hr at 30°C. Afterwards, cells were washed extensively in YEPD + pronase (100 µg/ml) and in YEPD to remove the alpha-factor. Cells were then grown in fresh YEPD at 30°C and protein extracts were prepared immediately and in intervals of 15 min using the method described in *Kushnirov, 2000*. For FACS staining, cells for each timepoint were fixed in 95% EtOH overnight at 4°C. Fixed cells were washed in 50 mM Tris-HCl pH 7.5 and sonicated using an ultrasonic waterbath. RNA was digested using 0.2 mg/ml RNAse (Sigma) for 2 hr and 50°C. Proteinase K (20 mg/ml pH 7.5; Roche) was added and cells were incubated further for 2–2.5 hr at 50°C (method described in *Haase and Reed, 2002*). Cell suspension was stained with 1 µM Sytox Green (Invitrogen) for 30 min at room temperature. Stained cells were stored at 4°C and read out using the MaxQuant VJB machine.

## Lambda phosphatase treatment

To quantify the signal intensity of Ame1 phosphorylation isoforms, strains that carry two-micron plasmids were overexpressed by adding galactose to the medium and whole-cell extracts were prepared as follows: cells were incubated overnight in YEPD and diluted back into YEP + RG medium to induce overexpression. After 5 hr in YEP + RG, a 50 ml logarithmic growing culture was pelleted, washed once in 1× PBS, and the pellet was taken up in lysis buffer (25 mM HEPES pH 8, 150 mM NaCl, 2 mM EDTA, 1 mM DTT, 5% glycerol, 1:100 protease inhibitors; EDTA-free tablets Pierce, Thermo Fisher Scientific). Lysis was generated through mechanic disruption using acid-washed glass beads and vortexing for 1.5 min at 4°C, 1 mM DTT in the lysis buffer, and additionally 0.1% NP-40. The supernatant was spun at high speed and protein concentration was measured using Bradford Reagent. 30 µl lysate with 2 mg/ml protein concentration was treated with 1 µl of lambda phosphatase (NEB) for 30 min at room temperature. Reaction was stopped by adding 6× loading dye and boiling. Phosphatase treatment was followed by western blotting.

## In vitro kinase reactions

For in vitro kinase reactions, purified AO complexes (wild-type or different phospho-mutants in Ame1) were phosphorylated using Cdc28-Clb2. AO with a concentration of 5 µM was incubated

with recombinant Cdc28-Clb2, 1 μl 1 mM cold ATP, 1 μl radioactive labeled ATP (10 μCi/μl), and 4× kinase buffer (80 mM HEPES pH 7,5, 400 mM KCl, 40 mM MgCl$_2$, 40 mM MnCl$_2$, 100 mM β-glycerol-phosphate, 4 mM DTT) in a final volume of 20 μl. The reaction was incubated for 30 min at 30°C and stopped by adding 6× SDS loading dye. The whole reaction was loaded on a SDS-PAGE and stained with Coomassie Brilliant Blue. After destaining the gel overnight, an X-ray film was exposed for 5–20 min and developed with CAWOMAT 200 IR machine.

For kinase assays with Ame1-Okp1 complexes and Mtw1c, recombinant proteins were preincubated in a molar ratio of 2:1, where always 10 μM of AO and 5 μM Mtw1c were used. Mixed protein samples were incubated on ice for 30 min, followed by adding all other components described above and starting the kinase reaction.

For the quantitative phosphorylation analysis presented in *Figure 1—figure supplement 1*, 1 μM AO complex was phosphorylated with 0.3 nM cyclin-Cdc28 complex in a buffer containing 50 mM HEPES-KOH, pH 7.4, 150 mM NaCl, 5 mM MgCl$_2$, 20 mM imidazole, 2% glycerol, 0.2 mg/ml BSA, 500 nM Cks1, and 500 μM ATP (with added [γ-$^{32}$P]-ATP [Hartmann Analytic]). The reaction was stopped at 10 min (initial velocity condition, less than 10% of initial substrate turnover) by addition of SDS loading dye. The samples loaded on SDS-PAGE and γ-$^{32}$P phosphorylation signals were detected using Amersham Typhoon 5 Biomolecular Imager (GE Healthcare Life Sciences).

## Computational details of molecular dynamics simulations

The crystal structure of Cdc4 with a phosphopeptide (PDB ID: 1NEX [*Orlicky et al., 2003*] resolution: 2.7 Å) was used for the initial coordinates of our models. The reported complex comprises Cdc4, Skp1, and a linker region. For the present study, we removed Skp1 and the linker region as both are not directly involved in peptide binding. Further, the ions and crystallization water molecules were removed. The selenomethionine modifications in the protein structure were replaced by the parent methionine residues.

The peptides under study were modeled using the PEP-FOLD3.5 webserver (*Lamiable et al., 2016*). For performing the simulations, the control peptide in the crystal structure was removed and manually replaced by the candidate peptides, with further optimization using the FlexPepDock webserver (*London et al., 2011*). All the simulations were performed using the AMBER software suite (*Case et al., 2014*). The protein counterpart and the regular amino acids in the peptide were simulated with the modified ff99SB AMBER force field (*Maier et al., 2015*). The phosphorylated serine and threonine parameters were taken from the literature (*Homeyer et al., 2006*). GaMD simulations (*Miao et al., 2015*) were performed in explicit solvent using a TIP3P water (*Jorgensen et al., 1983*) box with sufficient counterions (Na$^+$/Cl$^-$) to ensure the overall electroneutrality of the system. The parameters of the ions were taken from the literature (*Joung and Cheatham, 2008*). Periodic boundary conditions were implemented along with Particle Mesh Ewald (PME) for computing the long-range electrostatic interactions (*Cheatham et al., 1995*). The systems were minimized in two steps (using 10,000 conjugate gradient and 10,000 steepest descent cycles for both steps). In step 1, the protein-peptide complex was restrained using a force constant of 50 kcal/mol A$^{-2}$, and only the ions and solvent molecules were allowed to relax. In step 2, the restraints were removed, and the whole assembly was allowed to relax. The minimized systems were heated to room temperature (40 ps) and underwent equilibration (5 ns). Finally, GaMD simulations were performed for 100 ns (three replicas, NPT). The trajectories were processed using the Cpptraj code (*Roe and Cheatham, 2013*), and the downloadable version of the Ligplot tool was used to create 2D interaction plots (*Wallace et al., 1995*). For the clustering of the GaMD replicas, we utilized the concepts of hierarchical agglomerative clustering (*Shao et al., 2007*), with a cutoff distance of 3 Å between any identified clusters.

## Acknowledgements

We thank Mathias Peter and Sue Biggins for providing strains and plasmids. We also thank Maren Soldierer and Tabea Graszynski for initial experimental contributions and all members of the Westermann lab for discussions. This work received support from the German Research Foundation (DFG), grant no. WE-2886/2. ES-G was also supported by Germany´s Excellence Strategy – EXC 2033 – 390677874 – RESOLV, funded by the DFG. SW and ES-G received funding from the collaborative research center CRC 1093 'Supramolecular Chemistry on Proteins' (Subprojects A8 and B7), funded

by the DFG. SW and ES-G acknowledge support from the collaborative research center CRC 1430 'Molecular Mechanisms of cell state transitions,' funded by the DFG. ES-G acknowledges the computational time provided by the supercomputer magnitUDE of the University of Duisburg-Essen. The work was funded by ERC consolidator grant 649124, Centre of Excellence for 'Molecular Cell Technologies' TK143 and Estonian Science Agency grant PRG550 to ML.

## Additional information

### Funding

| Funder | Grant reference number | Author |
|---|---|---|
| Deutsche Forschungsgemeinschaft | WE-2886/2 | Miriam Böhm Stefan Westermann |
| Deutsche Forschungsgemeinschaft | CRC1093 | Elsa Sanchez-Garcia Stefan Westermann |
| Deutsche Forschungsgemeinschaft | CRC1430 | Elsa Sanchez-Garcia Stefan Westermann |
| H2020 European Research Council | 649124 | Mart Loog |
| Estonian Science Foundation | PRG550 | Mart Loog |
| University of Tartu | TK143 | Mart Loog |
| German Research Foundation | EXC 2033 – 390677874 | Elsa Sanchez-Garcia |

The funders had no role in study design, data collection and interpretation, or the decision to submit the work for publication.

### Author contributions

Miriam Böhm, Conceptualization, Investigation, Visualization, Writing - original draft; Kerstin Killinger, Alexander Dudziak, Investigation, Writing - review and editing; Pradeep Pant, Mihkel Örd, Formal analysis, Investigation, Writing - review and editing; Karolin Jänen, Simone Hohoff, Investigation; Karl Mechtler, Formal analysis, Investigation; Mart Loog, Elsa Sanchez-Garcia, Formal analysis, Supervision, Funding acquisition, Writing - review and editing; Stefan Westermann, Conceptualization, Formal analysis, Supervision, Funding acquisition, Writing - original draft

### Author ORCIDs

Miriam Böhm (iD) https://orcid.org/0000-0002-6054-1912
Kerstin Killinger (iD) https://orcid.org/0000-0002-1278-0302
Alexander Dudziak (iD) http://orcid.org/0000-0001-5082-3468
Pradeep Pant (iD) http://orcid.org/0000-0003-3890-1958
Elsa Sanchez-Garcia (iD) http://orcid.org/0000-0002-9211-5803
Stefan Westermann (iD) https://orcid.org/0000-0001-6921-9113

### Decision letter and Author response

Decision letter https://doi.org/10.7554/eLife.67390.sa1
Author response https://doi.org/10.7554/eLife.67390.sa2

## Additional files

### Supplementary files

• Transparent reporting form

### Data availability

All data generated or analysed during this study are included in the manuscript and supporting files.

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
