## [Decision Letter]

**Acceptance summary:**

This paper presents convincing genetic and biochemical evidence that the kinetochore protein Ame1(CENP-U) contains a phospho-degron which regulates its stability, and that phosphorylation of this degron is reduced when Ame1 is bound to another kinetochore protein, Mtw1(Mis12). These findings led the authors to an interesting model: they suggest that cell cycle-dependent phosphorylation of phosphodegrons in kinetochore proteins may allows cells to have an abundance of kinetochore proteins during kinetochore assembly in S-phase, ensuring efficient assembly, but avoid any excess kinetochore proteins during mitosis, when they could be detrimental. This is an attractive model for kinetochore biology that merits further exploration.

**Decision letter after peer review:**

Thank you for submitting your article "Differentially accessible Cdc4 phospho-degrons regulate Ctf19^CCAN^ kinetochore subunit stability in mitosis" for consideration by *eLife*. Your article has been reviewed by 3 peer reviewers, one of whom is a member of our Board of Reviewing Editors, and the evaluation has been overseen by Anna Akhmanova as the Senior Editor. The reviewers have opted to remain anonymous.

Essential revisions:

1) The conclusion that the binding of the Mtw1 complex shields the Ame1 phosphodegron is arguably one of the most significant and interesting claims made in this paper. However, the evidence presented to support this claim relies exclusively on in vitro data. Thus, this part is out of balance with other parts of the paper where some in vivo correlations are attempted/made. All reviewers felt that this finding requires additional confirmation in vivo.

This could take several forms. For example:

– Determining the phosphorylation status of Ame1 at kinetochores (or on chromatin) versus in the soluble fraction. This could be done in skp1-3 mutants in order to avoid degradation of the phosphorylated Ame1.

– Using mutants in which Mtw1c or Ame1 cannot bind to one another.

– Depletion of Mtw1c, measuring Ame1 amount and phosphorylation.

2) The simplification of the phosphorylation patterns of Ame1 to CPD-only and CPD-null for tracking throughout the cell cycle (Figure 6) is logical, but it would be good to see the data for WT Ame1.

3) Some additional evidence on the functional consequences of the phosphoregulation is required: does a failure to remove the phosphorylated pool indeed lead to mitotic defects, or does precocious phosphorylation/degradation lead to kinetochore assembly defects? Some experimental possibilities are listed below. We do not expect all to be addressed, but some evidence beyond of what is currently shown is needed.

– Figure 1 shows that expressing Ame1 phosphomutants has no impact on general viability. However, it would be interesting to characterize whether these mutants result in any modest mitotic phenotypes or cell cycle delays, including in a colony sectoring assay for mini-chromosome loss or other sensitized assays.

– If Ame1 is precociously degraded in S-phase, kinetochore assembly should be impaired. It may be possible to test this with the Ame1-CPD-ILTPP mutant, or by finding another way to induce precocious phosphorylation or degradation.

– The model predicts changes in Ame1 abundance throughout the cell cycle. Whether this is consistent with the data in Figure 6 is not entirely clear. Do the authors think that only a minor pool is affected? In the absence of clearer data, the authors should use caution in their wording throughout the paper.

– The model strongly predicts that the mitotic degradation of Ame1 does not impact its abundance at centromeres. Some measurement (fluorescence of a tagged Ame1 or a ChIP on centromere DNA) of Ame1 at centromeres before and through mitosis would help instill confidence in the proposal.

4) The reviewers suggest several modifications to the text that should make it easier to follow the authors' arguments:

4a) Discussion of the results in Figure 8:

The section 'overexpression of COMA is toxic to cells' reads a bit like an afterthought, or perhaps an attempt to address the critique that the paper focuses on Ame1 at the expense of other COMA/CCAN components. This section should either be experimentally strengthened or could be incorporated into previous sections. With respect to strengthening: the authors argue that expressing proteins of the COMA complex in excess could result in ectopic kinetochore assembly. If maintaining this claim, this should be assessed by looking at the localization of COMA proteins following overexpression.

The growth assay shows that overexpression of COMA is toxic, but this toxicity does not seem to be aggravated by making COMA non-phosphorylatable – suggesting that phosphorylation is largely irrelevant in the overexpression toxicity. Why the phosphomutants show less accumulation than the WT in this assay remained unclear.

4b) More consistent use in the Results section of a summary sentence stating the conclusions of the experiments described, prior to moving on to the next set of experiments, would make it much easier to follow the text.

The text as a whole could use copyediting, in particular a few commas need to be added to separate clauses and make sentences clearer.

4c) Citations:

Some citations in the introduction (particularly the review articles) are a bit dated, and more updated citations could be added instead of or in addition to what is already cited.

The discussion and citations could include papers that have touched on connections between SCF and the centromere/kinetochore in human cells, for example Davies/Kaplan JCB 2010 and Gascoigne/Cheeseman JCB 2013, and the results by Au/Basrai PLoS Genetics 2020 maybe deserve a little more discussion with respect to the influence of SCF on budding yeast mitosis. In line 1016, the paper cited may not be the one that the authors intended to cite.

4d) Title and abstract:

– The title does not reflect the central model. As it stands, it is about 'subunit' stability with no context or proposal for what this would accomplish for chromosome segregation to occur with high fidelity.

– Abstract lines 30-32 are hard to understand. The version of the same sentence in the introduction paragraph to the main text works better.

– Abstract lines 42/43: "…phospho-regulated clearance of excess CCAN subunits protects against ectopic kinetochore assembly and contributes to mitotic checkpoint silencing". This sentence would benefit from rewording for two reasons:

(i) The authors suggest that ectopic kinetochores may form if the CCAN phospho-regulation fails. This seems unlikely, since the excess kinetochore subunits will not have a binding platform on chromatin. However, an excess of soluble kinetochore proteins may compete for binding partners with the kinetochore-bound pool, and this may result in some mitotic defects.

(ii) It is unclear how the authors imagine that the observed phosphoregulation contributes to checkpoint silencing. The fact that SCF mutants arrest with an active checkpoint does not necessarily mean that the SCF or Ame1 degradation contributes actively to checkpoint silencing.

4e) Legends:

Generally, please provide more information in the legends. For example, in Figure 2, explain YEPD and YEPR; in Figure 4A/B mention that overexpression is used; in Figure 6A, explain how the different arrests were obtained. The diagram in the upper-right corner of Figure 1 appears to be intended as part of Figure 1a; this could be made clearer both visually and in the figure legend. In the legend to Supplementary Figure 2, A and B are reversed compared to the actual figure.

4f) Main text:

Line 65: the term 'inner centromere' is used for where the CCAN is located. 'Inner centromere' generally refers to the inter-kinetochore space on the chromatin between kinetochore-forming chromatin where the CPC is located in many species. I think 'inner kinetochore' is better to use here.

Lines 68-72: the wording suggests that Mif2/CENP-C and COMA make up the entirety of the CCAN. Reword to make it clear that you are merely highlighting these two components. As it is, this is likely to cause confusion later when you mention other CCAN subunits and list representatives of each major subcomplex (lines 126-127) without introducing these other components. Similarly, the introduction of the KMN network should more clearly explain the relationship between this network and the CCAN.

Lines 85-93: The introduction of the concept that CENP-A nucleosome assembly can lead to a new kinetochore is not clearly written. It starts with the idea of CENP-A overexpression in experiments and in the context of cancer. It then goes to tether-experiments of some CCAN components to chromosome arms sites in human cells, and then back to mechanisms in budding yeast for degrading overexpressed CENP-A. This is difficult to follow. In budding yeast, CEN DNA seems to drive the show. That being said, excess components not at centromeres would titrate out other components or otherwise poison faithful kinetochore assembly. The point that there is danger to kinetochores can be made without invoking ectopic kinetochore formation.

Line 87: one example of many in which the authors use a human gene name without the yeast equivalent, then soon thereafter (ex: line 93) use the yeast version. This should be standardized across the manuscript (or at least in the introduction/discussion) to avoid causing confusion.

Lines 130-136: The explanation for why the authors chose to pursue Ame1 phosphorylation and exclude other CCAN subunits with promising phosphorylation motifs (for example: Mcm21, Nkp1) is incomplete/insufficient.

Line 146 "Okp1 is a better substrate for.… Cdk1-Clb5" – this seems to depend on what one compares to. Compared to Ame1, yes, but even Okp1 seems a better substrate for Cdk1-Clb2 than for Cdk1-Clb5. Please re-phrase. The following sentence only mentions hp-dependence of Okp1, although Ame1 phosphorylation also seems to depend on the hp.

Line 184: The graph in Figure 2C lacks the quantification of the 7E mutant. Either only cite 2B, or add the 7E data to 2C.

Line 189: first example of a few in which a yeast mutant is given by name without explanation of what it represents. This example in particular is the first mention of mad1 in the text and so a reader who is unfamiliar with the field is unlikely to understand the relevance of the experiment presented. In some cases, better explanations of yeast strains are provided in the figure legends than in the text. A brief explanation of what mad1 is (in this example) would be very helpful to a non-expert reader.

Line 194: Please define the abbreviation AO-or avoid it altogether.

Lines 390-401: This paragraph is a bit hard to follow without the context provided in the last sentence. Perhaps it could be reorganized to make it easier for the reader to follow and understand the significance of the results described.

On line 422, the authors refer to Figure 3 a,b as showing stabilization of Ame1 in the absence of SCF. However, it is actually Figure 4 that shows this.

Line 443 "other kinases must contribute": There are likely other phosphorylation site, but it seems formally possible that those are CDK1 sites – albeit not fitting the consensus.

Line 519: "does not fully prevent SCF regulation of the inner kinetochore". It may be better to say "does not fully prevent the mitotic effects of SCF" or something along these lines, leaving out "inner kinetochore"-since SCF may well act on other mitotic structures than the inner kinetochore.

Table 2: The phosphoglycerate kinase antibody may not have been used in this paper?

(5) Other technical suggestions:

– Multiple phosphorylated forms of Ame1 exist and run differently on a gel, which may influence the protein level comparisons in Figure 2B/E and S2. At least WT and 7A from Figure 2B should be treated with phosphatase (to down-shift all wild-type Ame1) before comparing the levels on an immunoblot.

– Figure 8A shows conservation of the Mcm21 phosphorylation site in closely related yeast species. Are the Ame1 sites conserved as well? Please add this information.

– (Optional) Figure 1F: For the longest of the deletions (delta31-116, delta31-187), a sequence-unrelated, flexible spacer could be inserted in order to address whether these mutants fail to function because specific sequences in this region are now missing, or because they have been shortened.

---

## [Author Response]

Essential revisions:1) The conclusion that the binding of the Mtw1 complex shields the Ame1 phosphodegron is arguably one of the most significant and interesting claims made in this paper. However, the evidence presented to support this claim relies exclusively on in vitro data. Thus, this part is out of balance with other parts of the paper where some in vivo correlations are attempted/made. All reviewers felt that this finding requires additional confirmation in vivo.This could take several forms. For example:– Determining the phosphorylation status of Ame1 at kinetochores (or on chromatin) versus in the soluble fraction. This could be done in skp1-3 mutants in order to avoid degradation of the phosphorylated Ame1.– Using mutants in which Mtw1c or Ame1 cannot bind to one another.– Depletion of Mtw1c, measuring Ame1 amount and phosphorylation.

We have constructed several new yeast strains to address this important point. In particular, we have combined Ame1-3A and -4A mutants with Mtw1-FRB in an anchor-away strain, which allows removal of the Mtw1 complex from the nucleus and follow Ame1 phosphorylation specifically at the identified motifs 1 and 2. The results of the experiment are shown in the new Figure 7C-E. Upon removal of Mtw1c from the nucleus, phosphorylation at motif 1 and 2 gradually increases over time (**7**D). Using this anchor-away setting we also analyzed the stability of phospho-forms of Ame1 in presence or absence of the binding partner Mtw1c. Upon inhibition of protein synthesis by cycloheximide, slowly migrating forms of Ame1 disappeared when Mtw1 was removed from the nucleus, but persisted when Mtw1c was still present (**7**E). Taken together, these in-vivo results complement the in vitro analysis on degron shielding by the Mtw1 complex.

2) The simplification of the phosphorylation patterns of Ame1 to CPD-only and CPD-null for tracking throughout the cell cycle (Figure 6) is logical, but it would be good to see the data for WT Ame1.

As requested, we have included the cell cycle experiment also for Ame1-WT in the new Figure 6—figure supplement 1A. The blot shows a dynamic phosphorylation pattern with phosphorylated Ame1 forms accumulating and some of them disappearing around min 60. The complexity of the pattern justifies the use of the Ame1-3A mutant to deconvolve the phosphorylation events and specifically analyze motif 1 and 2 phosphorylation.

3) Some additional evidence on the functional consequences of the phosphoregulation is required: does a failure to remove the phosphorylated pool indeed lead to mitotic defects, or does precocious phosphorylation/degradation lead to kinetochore assembly defects? Some experimental possibilities are listed below. We do not expect all to be addressed, but some evidence beyond of what is currently shown is needed.– Figure 1 shows that expressing Ame1 phosphomutants has no impact on general viability. However, it would be interesting to characterize whether these mutants result in any modest mitotic phenotypes or cell cycle delays, including in a colony sectoring assay for mini-chromosome loss or other sensitized assays.

We have performed a colony sectoring assay (Hieter et al., Cell 1985) to analyze differences in chromosome segregation fidelity between the Ame1 mutants. We find a modest increase in mis-segregation for the Ame1-7A mutant versus wild-type, the *ctf19* deletion strain serves as a positive control in this experiment.

**Author response image 1. sa2fig1:** Chromosome transmission fidelity experiment: Transmission of a chromosome fragment containing the SUP11 gene is assayed (Hieter et al. , 1985). Loss of the fragment leads to red or red sectored colonies in an *ade2-1* strain background. The *ctf19* deletion mutant serves as a control.

We have performed additional cell cycle experiments with Ame1 phosphorylation mutants to see if we can detect mitotic delays. Generally, there are no strong delays detectable over a single cell cycle. We see a subtle delay when phosphorylation is prevented at motif 2 (S41 S45), based on Pds1 degradation kinetics and in the FACS plots (Author response image 2, C), although the difference is so modest it may still be within the range of experimental variation.

**Author response image 2. sa2fig2:** Western blot analysis of Ame1-3A (phosphorylation at S41/45 and S52/53 allowed) versus Ame1-5A1 (phosphorylation at S41 S45 prevented). B. Quantification of Pds1 degradation in both samples. C. Corresponding FACS analysis, 0 min at bottom.

We also noticed that in the strains with C-terminally tagged Ame1-6xFlag variants and Pds1-myc, we sometimes observe >2C DNA signals in FACS, which complicates the analysis. Because of the subtlety of the effect, we show this additional analysis only here. In the manuscript, we have added additional explanations to the discussion, why the described Ame1 phospho-null mutant do not show stronger phenotypes, and how additional degron sequences can be identified and analyzed in future studies.

– If Ame1 is precociously degraded in S-phase, kinetochore assembly should be impaired. It may be possible to test this with the Ame1-CPD-ILTPP mutant, or by finding another way to induce precocious phosphorylation or degradation.

We have characterized the Ame1-CPD^ILTPP^ mutant further by crossing it to selected inner KT mutants predicted to be impaired in kinetochore assembly. We show that Ame1-CPD^ILTPP^ aggravates the growth defect of *ctf19* deletion mutants at 20 °C and at low benomyl concentrations (new Figure 5D).

– The model predicts changes in Ame1 abundance throughout the cell cycle. Whether this is consistent with the data in Figure 6 is not entirely clear. Do the authors think that only a minor pool is affected? In the absence of clearer data, the authors should use caution in their wording throughout the paper.

We have adjusted the wording throughout the text and indeed think that only a fraction of Ame1 is subjected to the type of phospho-regulation described in this study. We have improved the Discussion section to better cover this point.

– The model strongly predicts that the mitotic degradation of Ame1 does not impact its abundance at centromeres. Some measurement (fluorescence of a tagged Ame1 or a ChIP on centromere DNA) of Ame1 at centromeres before and through mitosis would help instill confidence in the proposal.

We have constructed the respective strains yeast and quantified the fluorescent signal of Ame1-WT-GFP and Ame1-7A-GFP at kinetochore clusters in metaphase and anaphase cells (see Author response image 3). We find no significant differences in the level of Ame1 at kinetochores by this method, consistent with the idea, that the abundance at centromeres is not affected.

**Author response image 3. sa2fig3:** Live cell microscopy of Ame1-WT and Ame1-7A tagged with GFP at metaphase and anaphase kinetochore clusters. Scale bar: 5 mm. The intensity of 15-18 kinetochore clusters for each strain was quantified. P-values calculated from unpaired Student’s t-test.

4) The reviewers suggest several modifications to the text that should make it easier to follow the authors' arguments:4a) Discussion of the results in Figure 8:The section 'overexpression of COMA is toxic to cells' reads a bit like an afterthought, or perhaps an attempt to address the critique that the paper focuses on Ame1 at the expense of other COMA/CCAN components. This section should either be experimentally strengthened or could be incorporated into previous sections. With respect to strengthening: the authors argue that expressing proteins of the COMA complex in excess could result in ectopic kinetochore assembly. If maintaining this claim, this should be assessed by looking at the localization of COMA proteins following overexpression.The growth assay shows that overexpression of COMA is toxic, but this toxicity does not seem to be aggravated by making COMA non-phosphorylatable – suggesting that phosphorylation is largely irrelevant in the overexpression toxicity. Why the phosphomutants show less accumulation than the WT in this assay remained unclear.

We agree that this section felt somewhat disconnected from the rest of the paper. We have removed the original Figure 8 and the corresponding section from the revised manuscript and include the information on the toxicity of overexpressing the entire COMA complex now as a separate panel in the new Figure 4D instead. We feel that this experiment is necessary to demonstrate the overall importance of COMA level regulation.

4b) More consistent use in the Results section of a summary sentence stating the conclusions of the experiments described, prior to moving on to the next set of experiments, would make it much easier to follow the text.The text as a whole could use copyediting, in particular a few commas need to be added to separate clauses and make sentences clearer.

We have edited the Results section to improve readability.

4c) Citations:Some citations in the introduction (particularly the review articles) are a bit dated, and more updated citations could be added instead of or in addition to what is already cited.The discussion and citations could include papers that have touched on connections between SCF and the centromere/kinetochore in human cells, for example Davies/Kaplan JCB 2010 and Gascoigne/Cheeseman JCB 2013, and the results by Au/Basrai PLoS Genetics 2020 maybe deserve a little more discussion with respect to the influence of SCF on budding yeast mitosis. In line 1016, the paper cited may not be the one that the authors intended to cite.

Thank you for pointing this out. We have removed the “Daniel et al.” citation from line 1016. We have added the citations Kaplan JCB2010 and Gascoigne/Cheeseman 2013 to the Discussion section (line 519 ff.) We also updated some of the citations in the introduction (e.g. line 80 ff, line 90).

4d) Title and abstract:– The title does not reflect the central model. As it stands, it is about 'subunit' stability with no context or proposal for what this would accomplish for chromosome segregation to occur with high fidelity.

As suggested, we have used the sentence from the introduction in the revised abstract (line 30). We have rephrased the title to make it more specific for the findings regarding Ame1 and to avoid too general claims.

– Abstract lines 30-32 are hard to understand. The version of the same sentence in the introduction paragraph to the main text works better.– Abstract lines 42/43: "…phospho-regulated clearance of excess CCAN subunits protects against ectopic kinetochore assembly and contributes to mitotic checkpoint silencing". This sentence would benefit from rewording for two reasons:(i) The authors suggest that ectopic kinetochores may form if the CCAN phospho-regulation fails. This seems unlikely, since the excess kinetochore subunits will not have a binding platform on chromatin. However, an excess of soluble kinetochore proteins may compete for binding partners with the kinetochore-bound pool, and this may result in some mitotic defects.(ii) It is unclear how the authors imagine that the observed phosphoregulation contributes to checkpoint silencing. The fact that SCF mutants arrest with an active checkpoint does not necessarily mean that the SCF or Ame1 degradation contributes actively to checkpoint silencing.

We thank the reviewer for these valuable points. We note that excess inner kinetochore proteins that contain DNA or chromatin-binding domains could be more susceptible to induce ectopic kinetochore formation than excess outer kinetochore components. We fully agree that titrating away subunits from the kinetochore-bound pool by excess unbound subunits is another important problem. We now mention both of these possibilities in the revised discussion (line 478 ff). We have removed the point on checkpoint silencing from the revised abstract.

4e) Legends:Generally, please provide more information in the legends. For example, in Figure 2, explain YEPD and YEPR; in Figure 4A/B mention that overexpression is used; in Figure 6A, explain how the different arrests were obtained. The diagram in the upper-right corner of Figure 1 appears to be intended as part of Figure 1a; this could be made clearer both visually and in the figure legend. In the legend to Supplementary Figure 2, A and B are reversed compared to the actual figure.

We have corrected these points in the legends to Figures 2, 4 and 6 of the revised manuscript. We have corrected the numbering in the new Figure 2—figure supplement 1C and D.

4f) Main text:Line 65: the term 'inner centromere' is used for where the CCAN is located. 'Inner centromere' generally refers to the inter-kinetochore space on the chromatin between kinetochore-forming chromatin where the CPC is located in many species. I think 'inner kinetochore' is better to use here.

We have changed the wording and now use the term “inner kinetochore”

Lines 68-72: the wording suggests that Mif2/CENP-C and COMA make up the entirety of the CCAN. Reword to make it clear that you are merely highlighting these two components. As it is, this is likely to cause confusion later when you mention other CCAN subunits and list representatives of each major subcomplex (lines 126-127) without introducing these other components. Similarly, the introduction of the KMN network should more clearly explain the relationship between this network and the CCAN.

We have rephrased the respective section to improve readability.

Lines 85-93: The introduction of the concept that CENP-A nucleosome assembly can lead to a new kinetochore is not clearly written. It starts with the idea of CENP-A overexpression in experiments and in the context of cancer. It then goes to tether-experiments of some CCAN components to chromosome arms sites in human cells, and then back to mechanisms in budding yeast for degrading overexpressed CENP-A. This is difficult to follow. In budding yeast, CEN DNA seems to drive the show. That being said, excess components not at centromeres would titrate out other components or otherwise poison faithful kinetochore assembly. The point that there is danger to kinetochores can be made without invoking ectopic kinetochore formation.

We have re-organized this section starting at line 85, and have removed the CCAN tethering experiments to streamline the text.

Line 87: one example of many in which the authors use a human gene name without the yeast equivalent, then soon thereafter (ex: line 93) use the yeast version. This should be standardized across the manuscript (or at least in the introduction/discussion) to avoid causing confusion.

As standard, upon first mention we use the respective yeast name with the superscripted human/general name e.g Cse4^CENP-A^.

Lines 130-136: The explanation for why the authors chose to pursue Ame1 phosphorylation and exclude other CCAN subunits with promising phosphorylation motifs (for example: Mcm21, Nkp1) is incomplete/insufficient.

We have added an additional explanatory sentence, see line 136 f.

Line 146 "Okp1 is a better substrate for.… Cdk1-Clb5" – this seems to depend on what one compares to. Compared to Ame1, yes, but even Okp1 seems a better substrate for Cdk1-Clb2 than for Cdk1-Clb5. Please re-phrase. The following sentence only mentions hp-dependence of Okp1, although Ame1 phosphorylation also seems to depend on the hp.

We have rephrased this point in the revised manuscript to make our statement more precise (line 143 ff). The respective section now reads: “Quantitative phosphorylation analysis confirmed that in the Ame1-Okp1 complex, Clb2-Cdk1 preferentially phosphorylated Ame1, whereas Clb5-Cdk1 preferred Okp1. The phosphorylation of the Ame1-Okp1 complex was dependent on the hydrophobic patch, a known substrate docking region in Clb5 and Clb2 cyclins, and a slightly stronger docking potentiation was seen in case of Okp1.”

Line 184: The graph in Figure 2C lacks the quantification of the 7E mutant. Either only cite 2B, or add the 7E data to 2C.

We have changed this and now only cite 2B in the text.

Line 189: first example of a few in which a yeast mutant is given by name without explanation of what it represents. This example in particular is the first mention of mad1 in the text and so a reader who is unfamiliar with the field is unlikely to understand the relevance of the experiment presented. In some cases, better explanations of yeast strains are provided in the figure legends than in the text. A brief explanation of what mad1 is (in this example) would be very helpful to a non-expert reader.

We have now included an explanation for the *mad1* deletion (line 197 ff).

Line 194: Please define the abbreviation AO-or avoid it altogether.

We introduce this in the revised manuscript in line 139.

Lines 390-401: This paragraph is a bit hard to follow without the context provided in the last sentence. Perhaps it could be reorganized to make it easier for the reader to follow and understand the significance of the results described.

We agree and have reorganized this section to start with the description of the *skp1-3* allele at different temperatures and then describe the results of the experiment (see line 409 ff).

On line 422, the authors refer to Figure 3 a,b as showing stabilization of Ame1 in the absence of SCF. However, it is actually Figure 4 that shows this.

We thank the reviewer for spotting this mistake, we have removed this section on overexpression of COMA from the revised manuscript.

Line 443 "other kinases must contribute": There are likely other phosphorylation site, but it seems formally possible that those are CDK1 sites – albeit not fitting the consensus.

We agree, although the observation that the combination of Ame1-7A and Okp1-S26A fully prevents Cdk1 phosphorylation in vitro (Figure 2—figure supplement 2), makes this relatively unlikely. We have cut this section of the manuscript, but mention the possibility of non-consensus Cdk1 sites in the revised discussion. See line 532 ff.

Line 519: "does not fully prevent SCF regulation of the inner kinetochore". It may be better to say "does not fully prevent the mitotic effects of SCF" or something along these lines, leaving out "inner kinetochore"-since SCF may well act on other mitotic structures than the inner kinetochore.

We agree and have rephrased this point (line 526).

Table 2: The phosphoglycerate kinase antibody may not have been used in this paper?

We have removed this from the materials section.

(5) Other technical suggestions:– Multiple phosphorylated forms of Ame1 exist and run differently on a gel, which may influence the protein level comparisons in Figure 2B/E and S2. At least WT and 7A from Figure 2B should be treated with phosphatase (to down-shift all wild-type Ame1) before comparing the levels on an immunoblot.

We have added the requested lambda phosphatase experiment as new Figure 2—figure supplement 1 in the revised manuscript. Level differences between Ame1-WT and Ame1-7A are confirmed after phosphatase treatment.

– Figure 8A shows conservation of the Mcm21 phosphorylation site in closely related yeast species. Are the Ame1 sites conserved as well? Please add this information.

We detect a conservation of the Cdk phosphorylation sites, especially T31, S41 S45 and S52/53 only in the most closely related Saccharomycetes strains (see Author response image 4). In more distantly related yeasts such as Candida glabrata or Kluyveromyces lactis, they are not obviously conserved. We have added this information to the manuscript (line 152 f).

**Author response image 4. sa2fig4:** Multiple sequence alignment of Ame1 N-terminus from difference yeast species. Conserved residues colored according to ClustalW scheme.

– (Optional) Figure 1F: For the longest of the deletions (delta31-116, delta31-187), a sequence-unrelated, flexible spacer could be inserted in order to address whether these mutants fail to function because specific sequences in this region are now missing, or because they have been shortened.

Due to time constraints, we have not constructed these variants, yet. We have added a sentence to the result section acknowledging the different possibilities for the observed defects (line 174 ff).